# Acute EPA-induced learning and memory impairment in mice is prevented by DHA

Ji-Hong Liu[1], Qian Wang[1], Qiang-Long You[1], Ze-Lin Li[1], Neng-Yuan Hu[1], Yan Wang[2], Zeng-Lin Jin[3], Shu-Ji Li[1], Xiao-Wen Li[1], Jian-Ming Yang[1], Xin-Hong Zhu [1], Yi-Fan Dai[2], Jiang-Ping Xu[4], Xiao-Chun Bai [5] & Tian-Ming Gao [1✉]

Eicosapentaenoic acid (EPA), an omega-3 fatty acid, has been widely used to prevent cardiovascular disease (CVD) and treat brain diseases alone or in combination with docosahexaenoic acid (DHA). However, the impact of EPA and DHA supplementation on normal cognitive function and the molecular targets of EPA and DHA are still unknown. We show that acute administration of EPA impairs learning and memory and hippocampal LTP in adult and prepubescent mice. Similar deficits are duplicated by endogenously elevating EPA in the hippocampus in the transgenic fat-1 mouse. Furthermore, the damaging effects of EPA are mediated through enhancing GABAergic transmission via the 5-HT$_6$R. Interestingly, DHA can prevent EPA-induced impairments at a ratio of EPA to DHA similar to that in marine fish oil via the 5-HT$_{2C}$R. We conclude that EPA exhibits an unexpected detrimental impact on cognitive functions, suggesting that caution must be exercised in omega-3 fatty acid supplementation and the combination of EPA and DHA at a natural ratio is critical for learning and memory and synaptic plasticity.

[1] State Key Laboratory of Organ Failure Research, Key Laboratory of Mental Health of the Ministry of Education, Guangdong-Hong Kong-Macao Greater Bay Area Center for Brian Science and Brain-Inspired Intelligence, Guangdong Province Key Laboratory of Psychiatric Disorders, Department of Neurobiology, School of Basic Medical Sciences, Southern Medical University, 510515 Guangzhou, P.R. China. [2] State Key Laboratory of Reproductive Medicine and Jiangsu Key Laboratory of Xenotransplantation, Nanjing Medical University, 210029 Nanjing, P.R. China. [3] Beijing Institute of Pharmacology and Toxicology, 100850 Beijing, P.R. China. [4] Department of Pharmacology, School of Pharmaceutical Sciences, Southern Medical University, 510515 Guangzhou, P.R. China. [5] Department of Cell Biology, School of Basic Medical Sciences, Southern Medical University, 510515 Guangzhou, P.R. China. ✉email: tgao@smu.edu.cn

The brain is highly enriched with fatty acids, which regulate both the structure and the function of neurons, glial cells and endothelial cells[1]. Similar to essential amino acids, some fatty acids are essential for the human body because mammals cannot generate them de novo, relying instead on a constant supply of food[1]. Usually, these fatty acids are long-chain polyunsaturated fatty acids found in various plant and marine life as the precursors linoleic acid and αlinolenic acid and are metabolized by elongation and desaturation into omega-3 fatty acids, such as arachidonic acid (AA), EPA, and DHA in mammals[2]. The conversion of these precursors into omega-3 fatty acids is mostly hepatic, although other organs such as the brain express the necessary enzymatic machinery[2].

EPA and DHA are two of omega-3 fatty acids, and many lines of evidence indicate that supplementation with EPA and DHA is clinically useful for preventing CVD[3–5]. In addition to the beneficial effects of omega-3 fatty acids on peripheral organ disorders such as CVD, studies focusing on the brain have also found that supplementation with DHA and EPA can alleviate disease of the central nervous system. For example, a meta-analysis performed in 35 double-blind RCTs including 6665 participants receiving omega-3 HUFAs and 4373 participants receiving placebo found a positive role of omega-3 fatty acids supplementation on depression[6]. And several epidemiological or observational investigations on more than 20,000 subjects also reported that higher dietary intake of fish or n-3 PUFAs are associated with decreased risk of depressive disorders or fewer depressive symptoms. Moreover, they further found that, EPA, mostly at 1 or 2 g/day, is better than placebo and DHA as a monotherapy or adjuvant in the treatment of mild-to-moderate depression[7–12].

Besides, a case-control study in Norway found an inverse association between consumption of marine fish, which contains amount of the omega-3 fatty acids, and risk of developing multiple sclerosis[13]. Additionally, findings from clinical and observational studies suggest that omega-3 fatty acids have a beneficial effect against ischemic stroke[14], social anxiety disorders[1], autistic spectrum disorders[15], attention deficit/hyperactivity disorder (ADHD)[16,17] and other diseases. Also, fish oil that contains high amounts of omega-3 fatty acids has been reported to benefit for cognition[18,19]. Moreover, studies focusing on DHA have found that DHA supplementation can alleviate age-related cognitive decline[20–22]. However, little is known about the effect of EPA and DHA supplementation on normal cognitive function or the molecular target of omega-3 fatty acids in brain.

In this study, we firstly investigated the effect of EPA supplementation on learning and memory in vivo. Surprisingly, we found that acute administration of EPA at a series of concentrations derived from the FDA-recommended dosage for humans impairs learning and memory and synaptic plasticity in adult and adolescent mice, an effect that is mediated by serotonin 6 receptor [5-hydroxytryptamine6 (5-HT$_6$) receptor, 5-HT$_6$R] acting on GABAergic neurons. Importantly, combined administration of DHA at a natural ratio can ameliorate EPA-induced impairments via the 5-HT$_{2C}$R. Our data reveal an unanticipated impact of EPA on cognitive function in mice, suggesting that caution must be exercised in omega-3 fatty acid supplementation and the ratio of EPA/DHA is critical for learning and memory and synaptic plasticity.

## Results

**EPA administration impairs learning and memory**. Adult (postnatal day 60 [P60]) male C57BL/6J mice were intragastrically dosed with EPA at a series of concentrations, which were derived from the FDA-recommended dosage for humans[23] using the Meeh-Rubner formula (see Methods section). One hour after administration of EPA, we tested three hippocampus-dependent behavioral outputs: performance in the Morris water maze (MWM)[24], contextual fear conditioning[25] and novel object recognition (NOR)[26]. We found that compared to mice that did not receive EPA, EPA-treated mice exhibited impaired learning and memory, with poorer learning ability and memory performance in the MWM (Fig. 1a–e), a shorter freezing duration in contextual fear conditioning (Fig. 1f) and a lack of preference in NOR (Fig. 1g). In addition, we found no effect on the alternation rate or freezing time in the T-maze test and cued fear conditioning test, respectively (Supplementary Fig. 1a, b). The aforementioned behavioral changes did not result from changes in swimming speed, locomotor activity or sensory responses (Supplementary Fig. 1c–g). Gas chromatography (GC) results indicated that EPA treatment (50 mg/kg) significantly increased the level of that fatty acid in the hippocampus but not in the prefrontal cortex (PFC) or striatum (Fig. 1h), which may be due to hippocampal enrichment of fatty acid-binding protein 7 (FABP7) (Supplementary Fig. 1h), a protein that binds omega-3 fatty acids with high affinity and stores them[27]. However, we did not find a change in the level of other fatty acids (Supplementary Fig. 2a–c). Interestingly, the increased EPA level exhibited time dependence: it increased for only one to 2 h after EPA treatment (Supplementary Fig. 2d–g). Meanwhile, learning and memory behaviors were impaired at these time points when the EPA level was elevated, but when the level returned to normal, there were no lasting detrimental impacts on learning and memory (Supplementary Fig. 2h–n), suggesting that the impairment of learning and memory by EPA may persist only as long as there is an elevated level of EPA in the hippocampus. These results suggested that acute intragastric (i.g.) EPA impaired learning and memory.

To further explore the long-term effects of EPA, we administered EPA (50 mg/kg) intragastrically once a day for a period of one month and tested twenty-four hours after the last day of EPA administration. We found that EPA did not affect the fatty acids levels (Supplementary Fig. 3a–d) and learning and memory behaviors 24 h after the last day of i.g. administration of EPA (50 mg/kg) that lasted one month (Supplementary Fig. 3e–k), further supporting an acute impairment effect of EPA. To test the effects of EPA on prepubescent mice (P21), a developmental stage that may correspond to the 2nd and 3rd years of human life[28], we administered EPA (50 mg/kg) by the i.g. route and tested behavior one hour later. We found that acute i.g. administration of EPA also impaired learning and memory in prepubescent mice (Supplementary Fig. 4a–g).

**EPA treatment impairs hippocampal LTP**. Hippocampal long-term potentiation (LTP) is considered the cellular mechanism underlying learning and memory[29,30]. To investigate the cellular mechanism underlying EPA-induced behavioral impairment, we recorded field excitatory postsynaptic potentials (fEPSPs) from the dendritic region of the CA1 and compared the LTP induction between hippocampal slices taken from EPA-treated and control animals. We administered a series of concentrations of EPA to the mice. One hour later we sacrificed the mice and obtained the hippocampal slices immediately. After incubated in ACSF for another one hour, we recorded LTP and found that 50, 75, and 150 mg/kg EPA significantly suppressed high-frequency stimulation (HFS)-induced LTP in adult mice (Fig. 1i, j), as did EPA administration in prepubescent mice (Supplementary Fig. 4h, i). Moreover, EPA administration had no effects on basal synaptic transmission, as shown by input-output (I–O) curves (Supplementary Fig. 1i). To exclude possible effects of EPA metabolites on LTP, we directly added of EPA onto hippocampal slices at a series of concentrations based on the increased level in the

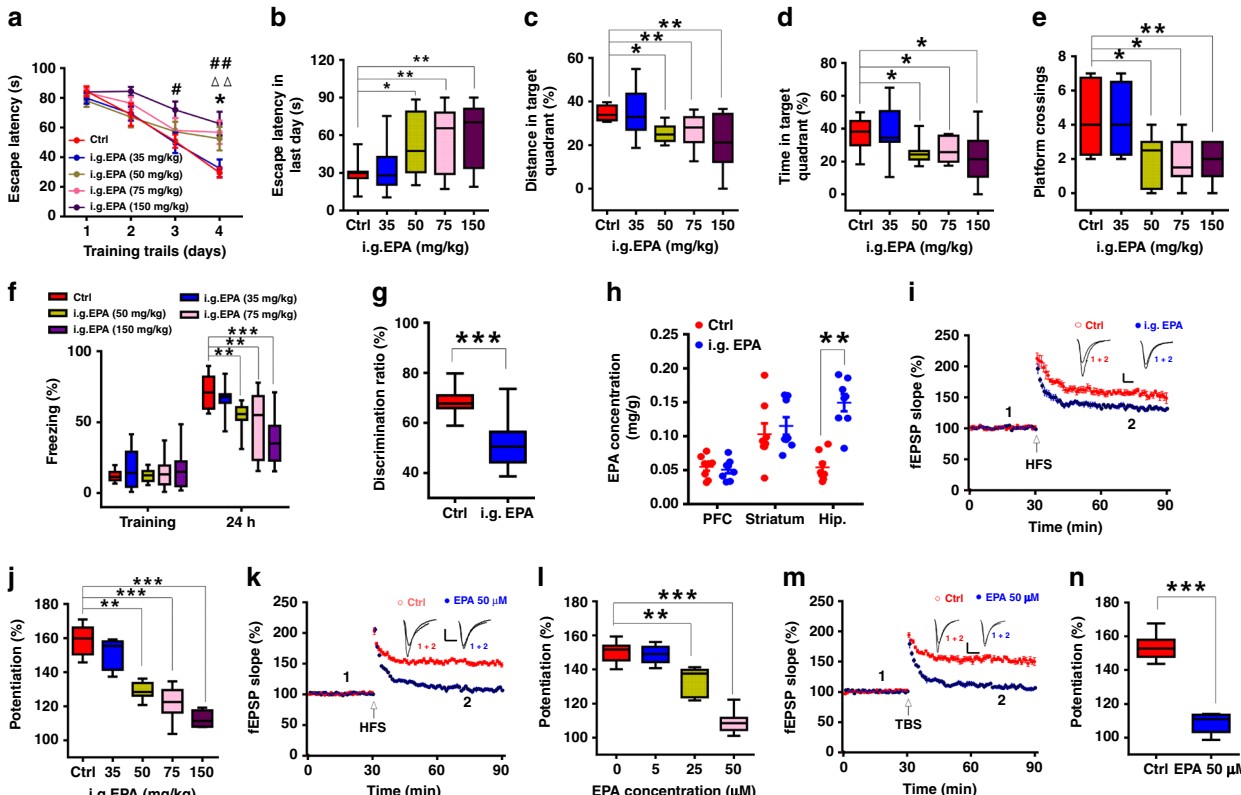

**Fig. 1 EPA administration impaired learning and memory and hippocampal LTP in adult mice (P60). a, b** Mean escape latencies across 4 consecutive days (**a**) or on the last day (**b**) during the MWM training (*n* = 9–11 mice/group; **a**: repeated measures two-way ANOVA, $F_{(4, 180)}$ = 55.518, *P* < 0.0001; **b**: one-way ANOVA, $F_{(4, 47)}$ = 54.568, *P* = 0.003; asterisk, triangle or hash indicates differences between EPA [50], EPA [75] or EPA [150] and Ctrl [saline-treated], respectively). **c, d** Percentage of distance (**c**) or time (**d**) spent in target quadrant during the probe trials (one-way ANOVA; **c**: $F_{(4, 47)}$ = 43.806, *P* = 0.016; **d**: $F_{(4, 47)}$ = 43.381, *P* = 0.018). **e** Number of platform crossings during the probe trials (one-way ANOVA; $F_{(4, 47)}$ = 45.431, *P* = 0.002). **f** Freezing time in the contextual fear conditioning test (*n* = 10–12 mice/group; one-way ANOVA; $F_{(4, 50)}$ = 48.636, *P* < 0.0001). **g** Discrimination ratio of time spent interacting with a novel object versus a familiar object in the NOR test (50 mg/kg, *n* = 10 mice/group; two-tailed Student's *t*-test; *P* < 0.0001). **h** I.g. administration of 50 mg/kg EPA increased the EPA level in the hippocampus (detected at 1 h after EPA administration; *n* = 8; two-tailed Student's *t*-test; *P* = 0.001). **i, j** I.g. administration of EPA suppressed HFS-induced LTP (recorded at 2 h after EPA administration; *n* = 5–7 slices/group; one-way ANOVA; $F_{(4, 26)}$ = 52.545, *P* < 0.0001). **k–n** Acute EPA treatment 10 min after establishing the baseline recording impaired HFS-LTP (**k, l**, *n* = 5–14 slices/group; one-way ANOVA; $F_{(3, 35)}$ = 47.807, *P* < 0.0001) and TBS-LTP (**m, n**, *n* = 7–8 slices/group; two-tailed Student's *t*-test; *P* < 0.0001). Scale bars: 0.5 mV, 5 ms. Data show mean ± s.e.m.. *\*P* < 0.05, *\*\*P* < 0.01, *\*\*\*P* < 0.001.

hippocampus after i.g. administration of 50 mg/kg EPA (Supplementary Fig. 1j). These ex vivo experiments showed that treatment with 25 or 50 μM but not 5 μM EPA similarly suppressed HFS-induced and theta burst stimulation (TBS)-induced LTP (Fig. 1k–n), which was consistent with the previous studies[31]. Taken together, these findings indicate that EPA impaired learning and memory-related behavior and LTP induction, both in vivo and ex vivo.

**Elevated EPA in the hippocampus contributes to the deficits in learning and memory and LTP in fat-1 mice.** To further explore the effect of EPA on learning and memory, we employed transgenic mice carrying the fat-1 gene, which encodes an n-3 fatty-acid desaturase enzyme that converts n-6 to n-3 fattyacids[32]. However, by GC analysis of samples from the brain, we found specific elevation of EPA in the PFC and hippocampus, with no changes in the levels of other fatty acids (Fig. 2a and Supplementary Fig. 5c–e); this finding implies that the model is appropriate for investigating the action of endogenously elevated EPA levels on learning and memory. We next examined hippocampus-dependent learning and memory in fat-1 mice. Behavioral experiments demonstrated that the fat-1 mice exhibited poorer performance than wild-type mice on these tests

(Fig. 2c–h). Further, electrophysiological recordings showed the failure of both HFS and TBS to induce LTP (Fig. 2i–l). However, there were no obvious differences in swimming speed, locomotor activity or basal synaptic transmission between fat-1 and control mice (Supplementary Fig. 5i–k). These results from transgenic fat-1 mice further support our hypothesis that EPA impairs learning and memory and synaptic plasticity.

To confirm whether these deficits in fat-1 mice were mediated by the increased level of EPA, we employed adeno-associated virus (AAV)-fat-1 short hairpin RNAs (shRNAs) to silence the fat-1 gene (Supplementary Fig. 5a, b). We found that hippocampal injection of AAV-fat-1 shRNA almost restored EPA levels to normal and rescued the deficits in behavioral tests (Fig. 2b–h) and LTP induction (Fig. 2i, j) in fat-1 mice, while the fat-1 mice injected with control shRNA still showed impaired behavioral performance and LTP (Fig. 2b–j). However, we did not observe differences in swimming speed or locomotor activity between fat-1 mice that were injected with the fat-1 shRNA and the control shRNA (Supplementary Fig. 5i, j); additionally, the levels of other fatty acids were similar in both groups (Supplementary Fig. 5f–h). These results further indicate that elevated EPA can lead to deficits in learning and memory and LTP.

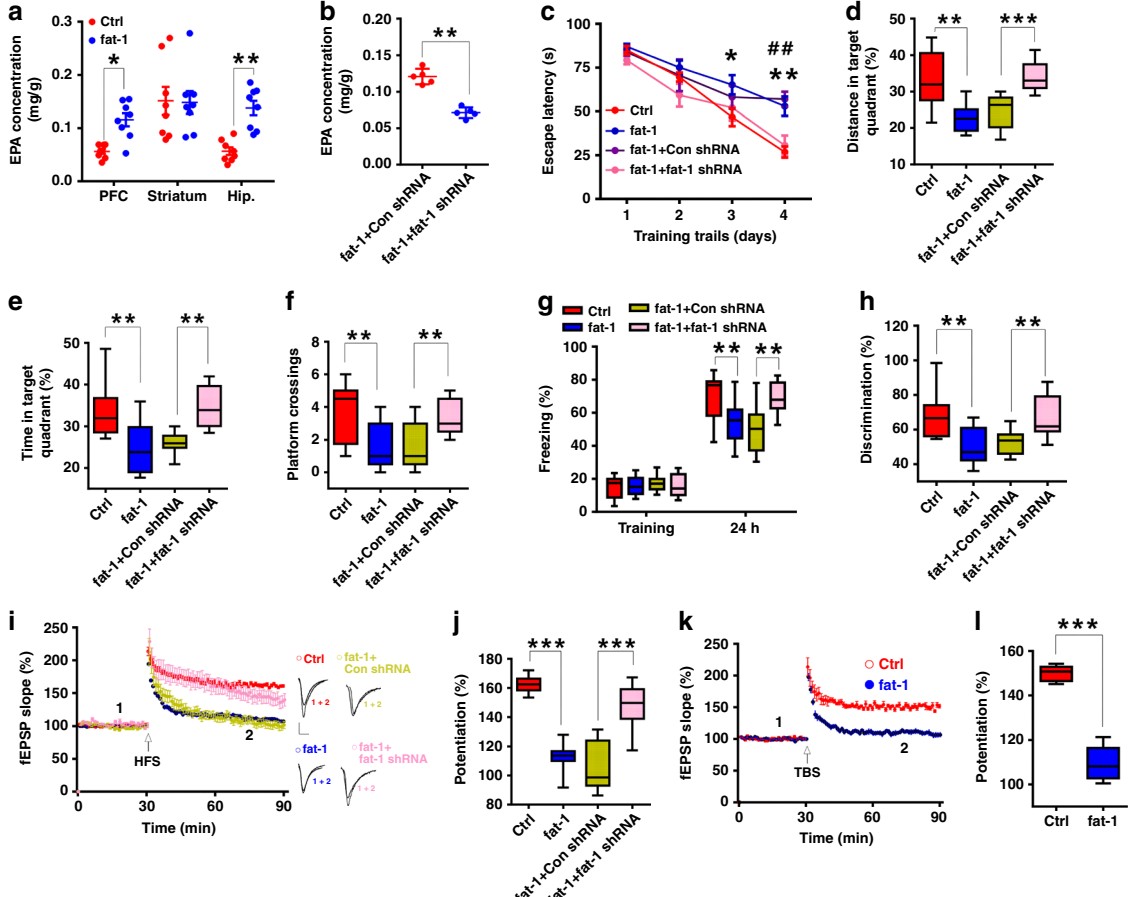

**Fig. 2 Elevated EPA in the hippocampus led to deficits in learning and memory and LTP in fat-1 mice. a** Fat-1 mice exhibited increased EPA level in the PFC and hippocampus ($n = 8$, two-tailed Student's $t$-test, PFC: $P = 0.036$; striatum: $P = 0.102$; Hip.: $P = 0.007$). **b** Effect of hippocampal knockdown of the fat-1 gene on the EPA level in fat-1 mice ($n = 4$–5, two-tailed Student's $t$-test, $P = 0.006$). **c**–**f** Performance in the MWM in fat-1 mice with or without intrahippocampal injection of shRNA ($n = 9$–10 mice/group; **c**: repeated measures two-way ANOVA, $F_{(3, 128)} = 45.215$, $P < 0.0001$; **d**: one-way ANOVA, $F_{(3, 33)} = 43.381$, $P = 0.013$; **e**: one-way ANOVA, $F_{(3, 33)} = 3.168$, $P = 0.012$; **f**: one-way ANOVA, $F_{(3, 33)} = 55.685$, $P = 0.010$). In **c**, asterisk or hash indicate differences between Ctrl and fat-1 groups, fat-1+Con shRNA and fat-1+fat-1 shRNA groups, respectively. **g**, **h** Freezing time in the contextual fear conditioning test ($n = 9$–10 mice/group; one-way ANOVA, $F_{(3, 33)} = 47.032$, $P = 0.002$) and discrimination ratio in the NOR test ($n = 9$–10 mice/group; one-way ANOVA, $F_{(3, 33)} = 53.126$, $P = 0.001$) after hippocampal knockdown of the fat-1 gene. **i**–**l** Suppressed HFS-LTP (**i**, **j**: Ctrl $n = 10$–26 slices/group; one-way ANOVA, $F_{(3, 59)} = 59.684$, $P < 0.0001$) and TBS-LTP (**k**, **l**: $n = 5$–6 slices/group; two-tailed Student's $t$-test, $P < 0.0001$) in hippocampal slices of fat-1 mice were rescued by fat-1 shRNA treatment. Data show mean ± s.e.m. Scale bars: 0.5 mV, 5 ms. *$P < 0.05$, **$P < 0.01$, ***$P < 0.001$.

**EPA has no effect on glutamatergic transmission**. The induction of LTP in the hippocampal CA1 region is mediated predominantly by glutamatergic synaptic transmission[33]. To investigate the mechanism underlying the EPA action on LTP, we first measured fEPSPs at the Schaffer collateral (SC)-CA1 synapse. However, we observed no detectable changes in basal synaptic transmission in terms of I–O curves or at baseline (Fig. 3a, b), and no detectable changes in the presynaptic release in paired-pulse facilitation (PPF) (Fig. 3c) after EPA treatment. Moreover, perfusion of the slices with EPA exhibited no effect on either the frequency or the amplitude of the spontaneous excitatory postsynaptic currents (sEPSCs) (Fig. 3d–f). We also added EPA following the induction of LTP to determine whether it would affect the expression of LTP, which is mainly mediated by alpha-amino-3-hydroxy-5-methyl-4-isoxazolepropionic acid (AMPA) receptor (AMPAR)[34]; we found that EPA did not suppress LTP in this condition (Fig. 3g, h). These results suggest that AMPAR-mediated synaptic transmission is not affected by EPA. To determine whether EPA regulates $N$-methyl-D-aspartate (NMDA) receptor (NMDAR)-mediated responses, we measured fEPSPs in the presence of 20 μM 6-cyano-7-nitroquinoxaline-2,3-

dione (CNQX) to block AMPAR and in $Mg^{2+}$-free buffer to release the NMDAR block. Interestingly, EPA had no effect on the slopes of NMDAR fEPSPs because the I–O curves with and without EPA completely overlapped in the two abovementioned conditions (Fig. 3i). We next measured NMDAR-mediated EPSCs in pyramidal neurons in a whole-cell configuration and observed no change in NMDAR-EPSCs after EPA treatment (Fig. 3j, k), in agreement with our results from fEPSPs recording. Together, these observations demonstrate that EPA does not alter glutamatergic transmission at hippocampal SC-CA1 synapses.

**5-HT₆ receptor mediates the impairing effect of EPA**. To further investigate the mechanism underlying the effect of EPA, we performed a radio-ligand receptor binding assay[35] and found that EPA potently inhibited the binding of [3H]-LSD to the 5-HT₆R (Supplementary Table 1), with an $IC_{50}$ of 26.62 μM (18.85–37.60 μM), suggesting that EPA may bind to 5-HT₆R. Since the serotonergic system is also implicated in the neurobiological control of learning and memory and synaptic plasticity[36–40], we studied the interaction between EPA and 5-HT₆R further by examining

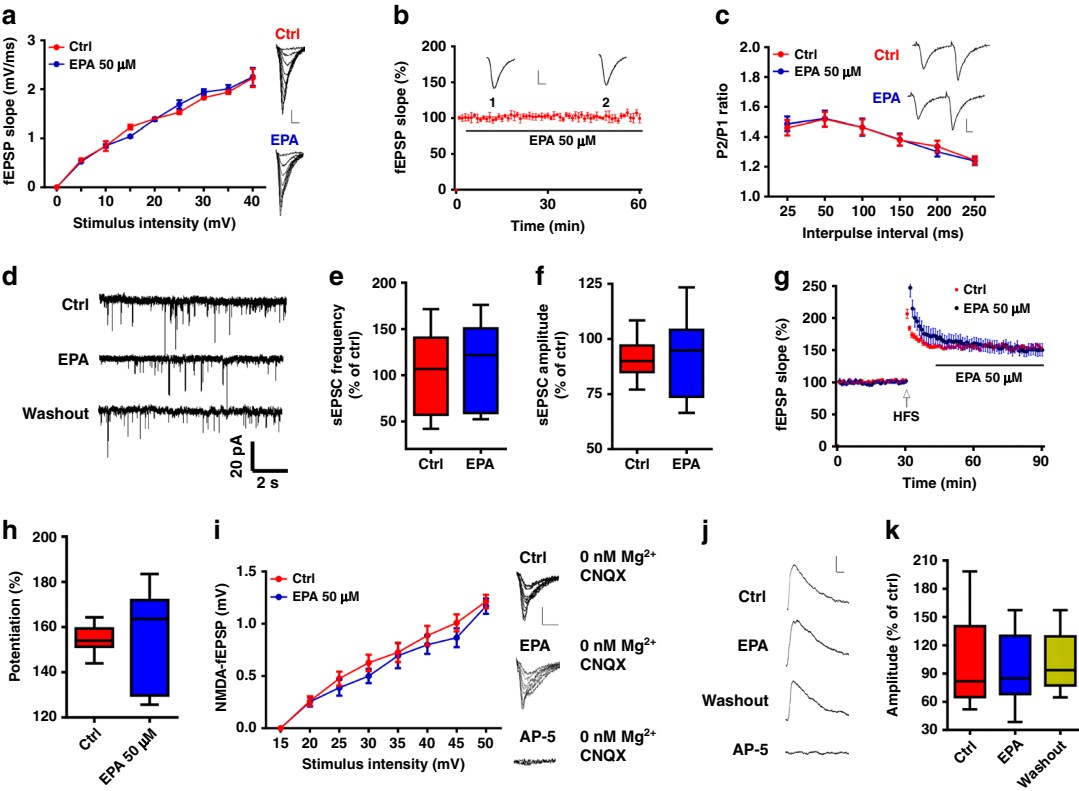

**Fig. 3 No effect of acute EPA (50 μM) treatment on glutamatergic transmission in hippocampal slices. a** I–O curves after acute EPA treatment ($n = 6$ slices/group; repeated measures two-way ANOVA, $F_{(1, 90)} = 18.367$, $P = 0.837$). **b** Bath application of EPA after establishing the baseline recording did not affect the fEPSP slope ($n = 5$ slices). **c** PPF after EPA treatment ($n = 6$ slices/group; repeated measures two-way ANOVA, $F_{(1, 60)} = 23.453$, $P = 0.812$); (right) representative recordings. Scale bars: 0.5 mV, 5 ms. **d–f** Effect of EPA on sEPSCs frequency (**e**, $n = 9$ slices, two-tailed Student's $t$-test, $P = 0.758$) and amplitude (**f**, two-tailed Student's $t$-test, $P = 0.938$). **g, h** Effect of EPA treatment on the LTP expression ($n = 7$–11 slices/group; two-tailed Student's $t$-test, $P = 0.833$). **i** NMDAR fEPSPs slopes after treatment with EPA ($n = 5$–6 slices/group; repeated measures two-way ANOVA, $F_{(1, 71)} = 22.739$, $P = 0.903$). fEPSPs were recorded in the presence of 20 μM CNQX and 0 nM Mg$^{2+}$. **j, k** Effect of EPA treatment on evoked NMDA currents ($n = 10$ cells; one-way ANOVA, $F_{(2, 27)} = 14.208$, $P = 0.678$). Scale bars: 100 pA, 10 ms. Data show mean ± s.e.m.

the effect of EPA on 5HT$_6$R function in 293T cells transfected with mouse 5-HT$_6$R plasmid. Enzyme-linked immunosorbent assay (ELISA) results indicated that, similar to the 5-HT$_6$R agonist EMD-368088 (EMD), EPA elevated the level of cyclic adenosine monophosphate (cAMP), a downstream target of the 5-HT$_6$R/adenylyl cyclase signaling pathway[41], which can be blocked by the 5-HT$_6$R antagonist SB-399885 (SB) (Fig. 4a). Moreover, this stimulation exhibited concentration dependence (Fig. 4b). Importantly, similar experiments using human 5-HT$_6$R plasmid showed the same results (Fig. 4c, d). We also confirmed that most regions of the brain including the hippocampus, expressed this receptor (Supplementary Fig. 6a). Moreover, the expression levels of 5-HT$_6$R in the hippocampus of prepubescent and adult mice showed no significant differences (Supplementary Fig. 6b). These observations suggest that EPA may act as a 5-HT$_6$R agonist in the brain.

To verify our hypothesis that EPA activates 5-HT$_6$R in the brain, we first employed the specific 5-HT$_6$R antagonist SB to test whether it could block the effect of EPA. On both training and testing day of contextual fear conditioning test, the antagonist was microinjected into the hippocampal CA1 region 30 min before i.g. administration of EPA (50 mg/kg), and one hour after EPA administration, the mice behaved similarly to control mice in terms of freezing duration; also, when pre-treated with SB, the EPA-treated mice behaved similarly to control mice in terms of discrimination in the NOR test after, while the antagonist alone did not have any effect (Fig. 4e, f). Moreover, we found that, when

pretreated with the 5-HT$_6$R antagonist SB, the hippocampal slices from the EPA-treated and control mice showed similar LTP (Fig. 4g, h). The ex vivo experiments also showed that 50 μM EPA was unable to impair hippocampal LTP when the slices were pretreated with the antagonist (Fig. 4i, j). Furthermore, the rescuing effect of the antagonist exhibited concentration dependence (Fig. 4j). However, the antagonist itself did not affect LTP or baseline synaptic transmission (Fig. 4j, k). In the transgenic fat-1 mice, we also found that the antagonist could rescue the impaired behavioral performance (Fig. 4l, m) and LTP (Fig. 4n, o). However, microinjection of the antagonist did not affect locomotor activity (Supplementary Fig. 7a,b).To test whether EPA exerted its effect by stimulating 5-HT release, we performed microdialysis experiments in i.g. EPA-treated mice and fat-1 mice and found that the elevated EPA level did not change the extracellular 5-HT concentration (Supplementary Fig. 1k,l). These results indicate that EPA may exert its detrimental effect by activating the 5-HT$_6$R.

To further confirm the relationship between EPA and 5-HT$_6$R in vivo, we used 5-HT$_6$R knockout (KO) mice. Western blotting and immunohistochemistry results verified the deletion of the receptor in the KO mice (Fig.5a–c). We then administered EPA (50 mg/kg) by the i.g. route and conducted behavioural tests. While KO mice not treated with EPA performed similarly to their control littermates[42], the impairing effect of EPA disappeared in EPA-treated KO mice in the MWM, contextual fear conditioning and NOR tests (Fig. 5d–i). Moreover, i.g. administration of EPA

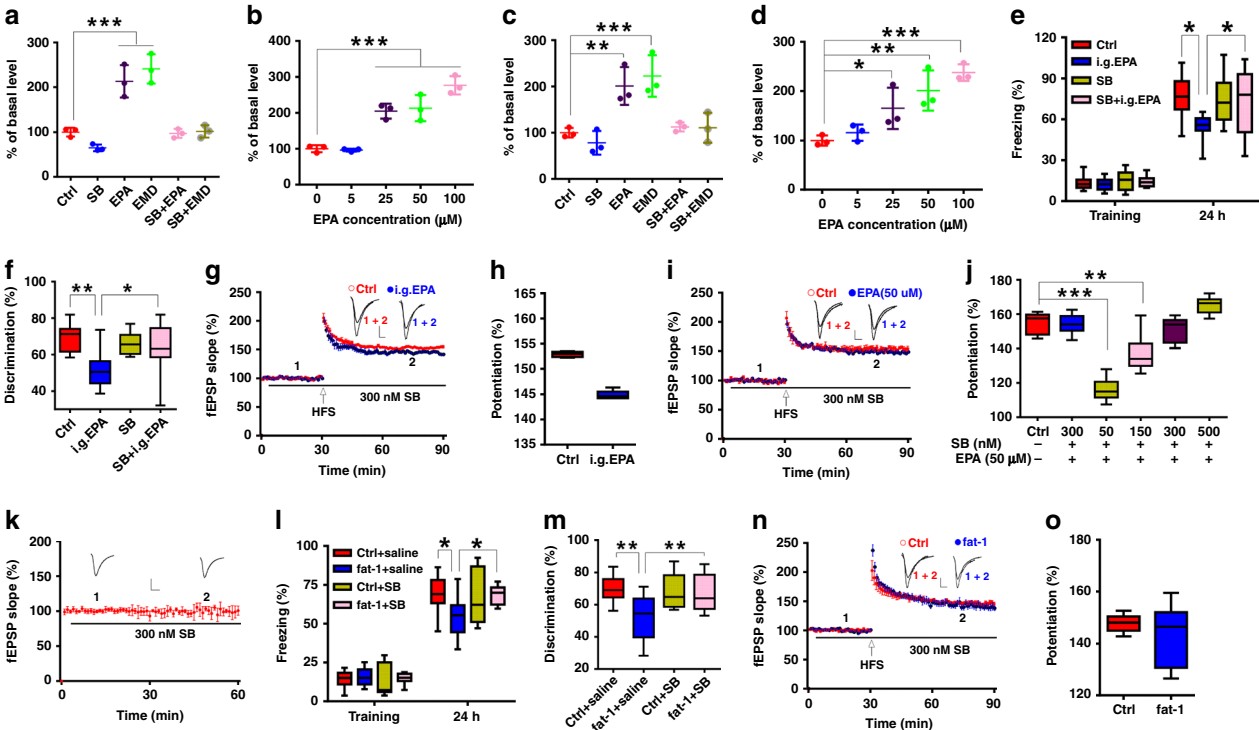

**Fig. 4 5-HT6R mediated the impairing effect of EPA on behavior and synaptic plasticity. a–d** EPA-stimulated cAMP production was mimicked by the 5-HT$_6$R agonist EMD-368088 (EMD,10 μM) but completely blocked by the 5-HT$_6$R antagonist SB-399885 (SB, 300 nM) in 293T cells transfected with the mouse (**a**, **b**) or human (**c**, **d**) 5-HT$_6$R plasmid ($n = 3$ wells/group; one-way ANOVA; **a**: $F_{(5, 12)} = 48.668$, $P < 0.0001$; **b**: $F_{(5, 12)} = 55.89$, $P < 0.0001$; **c**: $F_{(5, 12)} = 61.268$, $P < 0.0001$; **d**: $F_{(5, 10)} = 62.109$, P = 0.001). **e–h** The 5-HT$_6$R antagonist SB blocked the impairment in learning and memory (**e**: $n = 9$–11 mice/ group; one-way ANOVA, $F_{(4, 36)} = 43.560$, $P = 0.024$;(**f**): $n = 9$–10 mice/group; one-way ANOVA, $F_{(3, 35)} = 45.455$, $P = 0.003$) and LTP (**g**, **h**, $n = 5$ slices/group; two-tailed Student's $t$-test, $P = 0.518$) induced by i.g. administration of 50 mg/kg EPA. **i**, **j** SB prevented EPA-impaired LTP in a concentration-dependent manner ($n = 6$–11 slices/group; one-way ANOVA, $F_{(5, 40)} = 50.89$, $P < 0.0001$). **k** Bath application of SB after establishing the baseline recording did not affect fEPSP slope ($n = 5$). **l–o** CA1 microinjection of the 5-HT$_6$R antagonist SB (300 nM) rescued the impairments in behaviors in the contextual fear conditioning (**l**, $n = 8$–12 mice/group; one-way ANOVA, $F_{(3, 33)} = 42.182$, $P = 0.019$) and NOR test (**m**, $n = 8$–10 mice/group; one-way ANOVA, $F_{(3, 32)} = 44.931$, $P = 0.006$) and in LTP (**n**, **o**, $n = 6$ slices/group; two-tailed Student's $t$-test, $P = 0.877$) in fat-1 mice. Data show mean ± s.e. m. Scale bars: 0.5 mV, 5 ms. *$P < 0.05$, **$P < 0.01$, ***$P < 0.001$.

in the KO mice did not affect swimming speed in the MWM or locomotor activity in the open field test (Supplementary Fig. 7c, d). Furthermore, EPA treatment no longer suppressed LTP in the KO mice (Fig. 5j, k), while LTP in the KO mice without EPA treatment was intact. Thus, the effect of EPA on learning and memory is mediated by 5-HT$_6$R.

**EPA regulates GABAergic transmission via 5-HT$_6$R.** We next explored the mechanisms underlying the EPA-mediated impairment via 5-HT$_6$R. It has been reported that 5-HT$_6$R may be expressed in both glutamatergic and GABAergic neurons in the hippocampus[36]. The aforementioned results, showing no effect of EPA on glutamatergic transmission, suggest the involvement of GABAergic interneurons. To determine the localization of 5-HT$_6$R in GABAergic neurons, we used a specific antibody against 5-HT$_6$R to stain hippocampal sections from the green fluorescent protein (GFP)-expressing inhibitory neuron (GIN) mouse line, which expresses GFP under the control of a promoter (Gad1) that directs specific expression in GABAergic interneurons. As shown in Fig. 6a, 5-HT$_6$R was detected in GFP-positive cells, suggesting its expression in GABAergic interneurons. Next, we tested whether EPA could modulate GABA-mediated synaptic responses. Perfusion of the slices with 50 μM EPA led to an increase in the frequency of spontaneous inhibitory postsynaptic currents (sIPSCs) without affecting the amplitude (Fig. 6b–d). Moreover,

EPA also increased the feedforward IPSCs evoked by the HFS used to induce LTP (Fig. 6e, f), suggesting that EPA impaired LTP by enhancing GABAergic transmission. The increased sIPSC were blocked by the 5-HT$_6$R antagonist SB (Fig. 6g–i). Additionally, EPA failed to increase the frequency of sIPSCs in 5-HT$_6$R KO mice compared with their wild-type (WT) littermates (Fig. 6j, k). Moreover, ex vivo experiments showed that the GABA$_A$ receptor antagonist bicuculline methiodide (BMI) itself potentiated LTP only at higher concentration (20 μM) but not at lower concentration (<10 μM), while 10 μM BMI can prevent the EPA-induced LTP impairment (Fig. 6l, m), supporting an involvement of GABAergic mechanism in the EPA-impaired LTP. We also used the specific 5-HT$_6$R agonist EMD to mimic the EPA-induced effect and found that this agonist could also impair LTP (Fig. 6m). Similar to the effect of EPA and 5-HT$_6$R agonist EMD, we also observed that the GABA$_A$ receptor agonist diazepam impaired LTP (Fig. 6m). These observations suggest an important role of 5-HT$_6$R in the EPA-mediated regulation of GABAergic transmission.

To further test our hypothesis that EPA regulates GABAergic transmission via 5-HT$_6$R, we next employed AAV-5-HT$_6$R shRNAs to silence endogenous 5-HT$_6$R in GABAergic interneurons in the hippocampal CA1 region (Fig. 7a–e). We found that knockdown of 5-HT$_6$R prevented the impairing effect of EPA on GABAergic transmission (Fig. 7f, g), LTP (Fig. 7h–k) and animal behaviors (Fig. 7l–p) without affecting swimming speed in

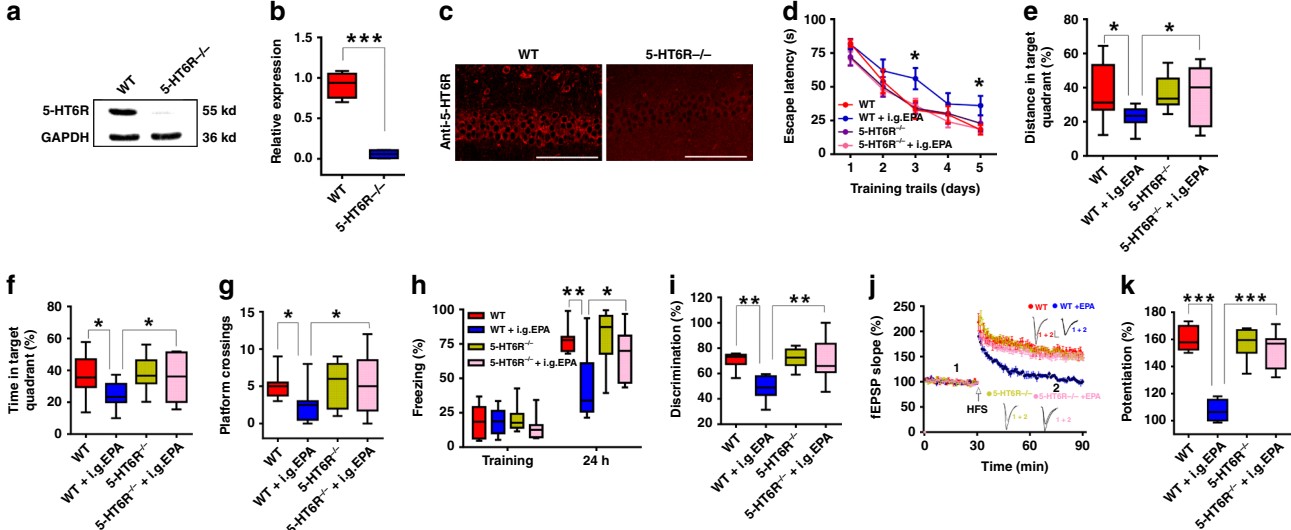

**Fig. 5 The impairing effect of EPA disappeared in 5-HT6R KO mice. a, b** Western blots of $5\text{-}HT_6R$ in the hippocampus from $5\text{-}HT_6R$ KO ($5\text{-}HT_6R^{-/-}$) mice and their WT ($5\text{-}HT_6R^{+/+}$) littermates ($n = 3\text{-}4$ experiments/group; two-tailed Student's $t$-test; $P < 0.0001$). Full-length blots are presented in Supplementary Fig. 9. **c** Specificity characterization of the anti-5-HT$_6$R antibodies. Scale bars: 100 μm. **d–k** The impairing effect of EPA on learning and memory in the MWM (**d–g**: $n = 10\text{-}12$ mice/group; **d**: repeated measures two-way ANOVA, $F_{(3,\ 200)} = 41.279$, $P = 0.053$; **e**: one-way ANOVA, $F_{(3,\ 39)} = 43.379$, $P = 0.028$; **f**: one-way ANOVA, $F_{(3,\ 39)} = 41.662$, $P = 0.02$; **g**: one-way ANOVA, $F_{(3,\ 39)} = 42.726$, $P = 0.057$), contextual fear conditioning (**h**: $n = 9\text{-}12$ mice/group; one-way ANOVA, $F_{(3,\ 38)} = 46.588$, $P = 0.001$), NOR (**i**: $n = 9\text{-}11$ mice/group; one-way ANOVA, $F_{(3,\ 35)} = 34.161$, $P = 0.005$), and LTP (**j**, **k**, $n = 5\text{-}8$ slices/group; one-way ANOVA, $F_{(3,\ 24)} = 38.908$, $P < 0.0001$) disappeared in 5-HT6R KO mice. Data show mean ± s.e.m. Scale bars: 0.5 mV, 5 ms. *$P < 0.05$, **$P < 0.01$, ***$P < 0.001$.

the MWM or locomotor activity in the open field test (Supplementary Fig. 7e, f).

**DHA prevents EPA-induced impairments via the 5-HT$_{2C}$R.** Marine fish oil, a rich source of omega-3 fatty acids, especially EPA and DHA, has no reported adverse effects on learning or memory; the same is true of combined supplementation with EPA and DHA[43,44]. An analysis of the composition of commonly consumed marine fish oils or EPA and DHA supplements in various representative experiments showed that the ratios of EPA to DHA were in the range of 1:1~1:2[45,46], suggesting that the ratio of EPA to DHA is critical for effects on learning and memory and other physiological processes. To address this possibility, we first examined the action of DHA on learning and memory and LTP and found that neither i.g. DHA at 50, 150 or 300 mg/kg (Supplementary Fig. 8) nor acute DHA at 100 μM treatment had any effect (Fig. 8h); these observations were consistent with previous studies[47]. We then tested the effects of combined EPA/DHA supplementation at different ratios on behavioral manifestations of learning and memory. We found that, similar to EPA alone, EPA (at a fixed dose of 50 mg/kg) and DHA at a ratio of 2:1 still impaired learning and memory compared with that of control mice, but this impairment disappeared at ratios of 1:1 and 1:2 (Fig. 8a–f). These treatments did not affect the swimming speed or locomotor activity of the mice (Supplementary Fig. 7g,h). In the LTP measurements, we found that the combined application of EPA (with a fixed concentration of 50 μM) and DHA at a ratio of 1:1 or 1:2 had no adverse effect on LTP, while the 2:1 combination still suppressed LTP (Fig. 8g, h). Additionally, EPA no longer increased the frequency of sIPSCs when combined with DHA at a ratio of 1:1 or 1:2 (Fig. 8i, j). Moreover, a similar preventive effect of DHA was observed on IPSCs evoked by HFS (Fig. 8k). In contrast to the EPA-induced enhancement of sIPSCs, perfusion of the slices with DHA alone led to a decrease in the frequency of sIPSCs without affecting the amplitude (Fig. 8i, j). These results suggest that combining EPA with DHA at a natural

ratio may prevent EPA from impairing learning and memory and LTP by balancing GABAergic transmission. To investigate the molecular target that DHA acts on, we performed a radio-ligand receptor binding assay and found that DHA potently inhibited the binding of [3H]-LSD to the 5-HT$_{2C}$R (Supplementary Table 1). In addition, ex vivo experiments showed that the 5-HT$_{2C}$R antagonist RS-102221 (RS) blocked the effects of DHA on the frequency of sIPSCs (Fig. 8l, m), while the antagonist itself did not affect GABAergic transmission (Fig. 8l, m). Moreover, EPA and DHA at a ratio of 2:1 was unable to restore the EPA-impaired sIPSCs and LTP when the slices were pretreated with the 5-HT$_{2C}$R antagonist RS (Fig. 8n–q). These observations suggest an important role of 5-HT$_{2C}$R in the DHA regulation of GABAergic transmission and LTP.

## Discussion
Omega-3 fatty acid molecules are the building blocks of the CNS. In contrast to other tissues, CNS tissue is enriched with omega-3 fatty acids. These fatty acids are indispensable to the normal development and function of the CNS[48]. αlinolenic acid (the precursors of EPA and DHA) and linolenic acid are not synthesized de novo by mammals, and a balanced diet containing appropriate amounts of these precursors is necessary to maintain sufficient brain levels of EPA and DHA[2,48]. Brain DHA decrease is associated with impaired cognitive ability and abnormal emotion[49]. Additionally, supplementation with DHA and EPA has been reported to be effective against the range of neurological and psychiatric disorders mentioned above. Therefore, it is accepted that omega-3 fatty acids are beneficial to the human body.

Nevertheless, in contrast to these beneficial effects, our results are, to our knowledge, the first to demonstrate that EPA is detrimental to learning and memory and synaptic plasticity in adult mice (P60), when given in an acute manner. Importantly, we also found that in prepubescent mice (P21), whose brain are developmentally similar to a 2- to 3-year-old human's brain[28], EPA administration impairs learning and memory. It is well known

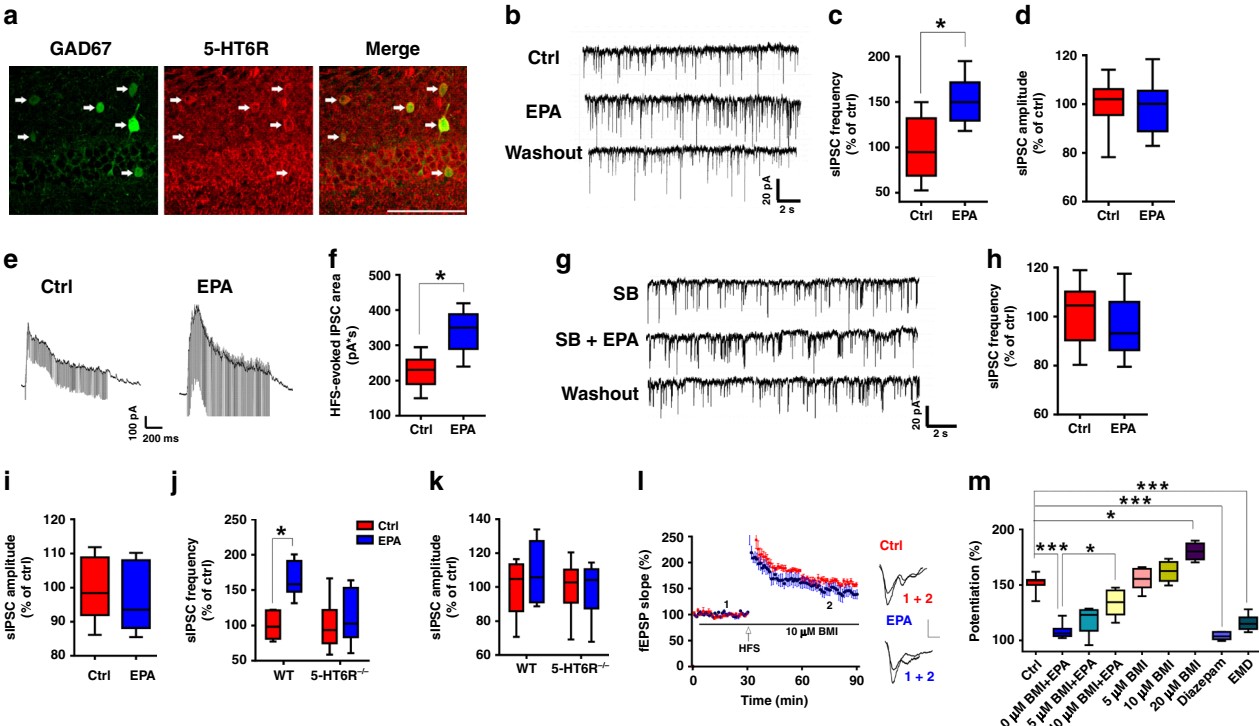

**Fig. 6 EPA modulated GABAergic transmission via the 5-HT$_6$R. a** Coronal sections of the hippocampal CA1 of GIN-GFP mice were stained with anti-5-HT$_6$R antibody. The arrows indicate neurons double positive for both 5-HT$_6$R and GAD67. Scale bar: 100 μm. **b–d** Effects of EPA on sIPSCs frequency and amplitude ($n = 9$ cells from five mice; two-tailed Student's $t$-test, **c**, $P = 0.013$; **d**, $P = 0.85$). Scale bars: 20 pA, 2 s. **e, f** HFS-evoked IPSCs in hippocampal CA1 neurons with or without EPA treatment ($n = 9$–11 cells from four mice; two-tailed Student's $t$-test, $P = 0.016$). **g–i** EPA was unable to enhance sIPSCs in the presence of the 5-HT$_6$R antagonist ($n = 10$ cells; two-tailed Student's $t$-test; **f**, $P = 0.868$; **g** $P = 0.798$).Scale bars: 20 pA, 2 s. **j, k** Increased frequency of sIPSCs induced by EPA disappeared in 5-HT$_6$R KO mice. The average sIPSC frequency is shown in **h** (two-tailed Student's $t$-test; WT, $P = 0.043$ and KO, $P = 0.604$), and the amplitude is shown in **i** (two-tailed Student's $t$-test; WT, $P = 0.546$ and KO, $P = 0.907$) ($n = 9$ cells from four mice/group). **l, m** The 5-HT$_6$R agonist EMD and the GABA$_A$ receptor agonist diazepam mimicked the impairing effect of EPA on LTP, but the GABA$_A$ receptor antagonist BMI at 10 μM can block the EPA effect on LTP ($n = 5$–11 slices/group; one-way ANOVA, $F_{(8, 47)} = 40.19$, $P = 0.021$). Data show mean ± s.e.m. *$P < 0.05$, ***$P < 0.001$.

that the infant formula available on the market contains percentages of EPA. Therefore, our results indicate that caution is necessary in the use of omega-3 fatty acids supplementation. In contrast, recent meta-analyses of clinical studies identified beneficial effects for treatment with EPA-rich formulations in the domains of long-term memory, working memory and problem solving[43]. However, these studies were performed in a chronic manner but there was a lack of acute effect of EPA on human cognition. So future investigations should take the time window of cognition detection into consideration.

Although the functional role of omega-3 fatty acids and the underlying mechanisms have been widely investigated in both peripheral tissues and central nervous system[1], the molecular target that EPA directly binds to and acts on is completely unknown. Glutamatergic synaptic transmission is reported to be the main modulator in the induction of LTP[29,30]. However, in evaluating the AMPAR- and NMDAR-mediated synaptic transmission, we did not observe a difference between the EPA-treated and control groups. This result excludes the hypothesis that EPA alters glutamatergic transmission at hippocampal SC-CA1 synapses. Meanwhile, in a radio-ligand receptor binding assay to screen the potential molecular targets underlying the effect of EPA, we found that 5-HT$_6$R is physically suited to be a specific binding site of EPA. Interestingly, EPA can accumulate the cAMP, the second messenger of the 5-HT$_6$R receptor[41] in 293T cells transfected with a mouse 5-HT$_6$R plasmid, which implies that EPA may be an endogenous 5-HT$_6$R agonist in the

brain. Most importantly, a similar effect occurs when a human 5-HT$_6$R plasmid is used, which supports the idea that EPA activates the receptor not only in animals but also in the human body. By using pharmacological and genetic manipulations to inhibit the function of 5-HT$_6$R, we found that EPA could no longer impair learning and memory or LTP induction in this condition. However, one possibility must be noted: EPA may also stimulate the release of 5-HT and then activate the receptor without being a direct ligand of 5-HT$_6$R. Indeed, EPA does not increase the 5-HT concentration in the hippocampus of mice treated intragastrically with EPA or that of fat-1 mice. Thus, our results firstly indicate that EPA may exert its detrimental effect by activating 5-HT$_6$R, which was supported by a number of clinical investigations confirming a direct relationship between 5-HT$_6$R and learning and memory[50–52].

Despite the wealth of information obtained from behavioral models, considerably less is known about the mechanism of action of 5-HT$_6$R ligands in the CNS. Recently, it has been reported that chronic treatment with 5-HT$_6$R antagonists increase the polysialylated form of neural cell adhesion molecule (PSA-NCAM), an effect that is positively correlated with cognitive function and enhanced synaptic plasticity[53]. In vivo microdialysis study has revealed that 5-HT$_6$R agonists can elicit robust elevations in extracellular concentrations of GABA in the dorsal hippocampus without affecting the release of norepinephrine, serotonin, dopamine, or glutamate[54]. Consistent with the microdialysis study, we observed here that EPA treatment had no

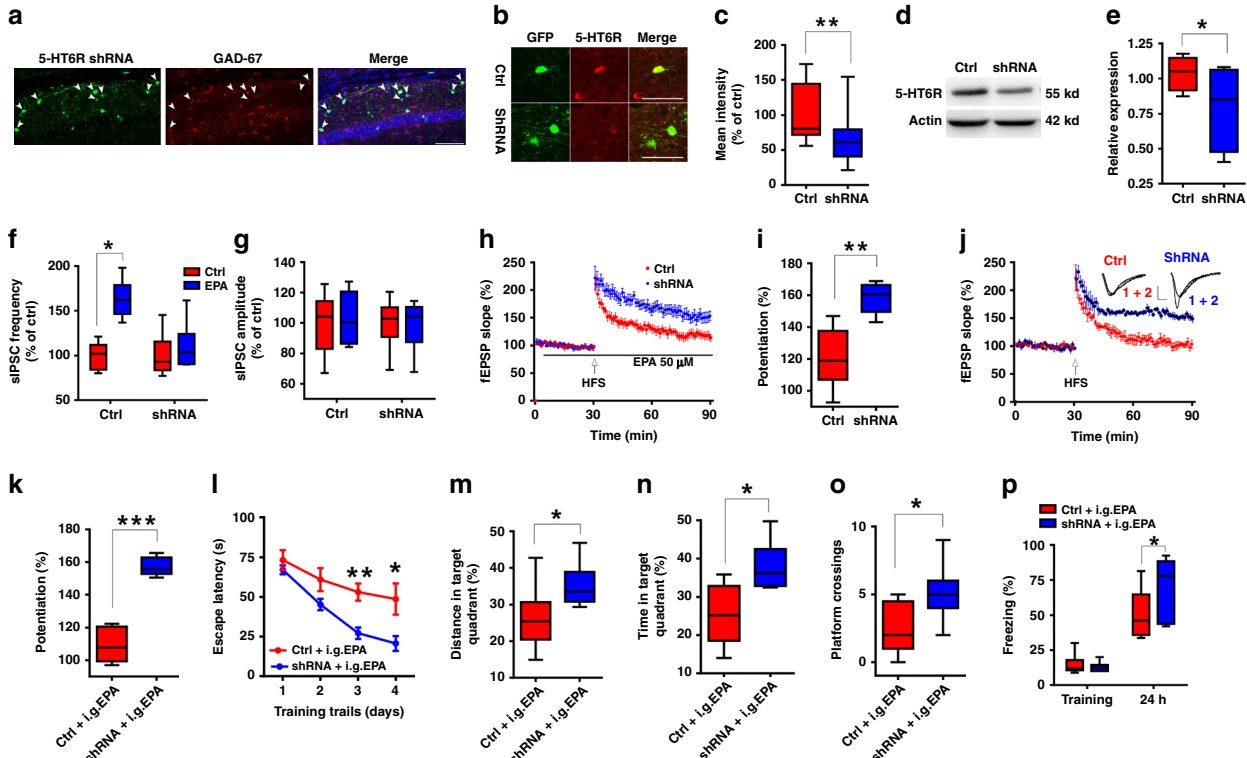

**Fig. 7 EPA impaired behaviors and synaptic plasticity via the 5-HT₆R on GABAergic interneurons. a** Representative fluorescence images showing most of the cells infected with pAAV-CAG-DIO-5-HT₆R-EGFP (shRNA) vectors are GABAergic neurons in the CA1 region of GAD-Cre mice. Scale bar: 100 μm. **b** Representative fluorescence images showing the knockdown of 5-HT₆R in CA1 GABAergic neurons infected with pAAV-CAG-DIO-5-HT₆R-EGFP (shRNA) vectors in GAD-Cre mice. Scale bar: 50 μm. **c** Histogram showing average fluorescence intensity (red) in the CA1 neurons from GAD-Cre mice that were injected with pAAV-CAG-DIO-EGFP (control) and shRNA virus (Ctrl group: $n = 20$ cells from four mice; shRNA group: $n = 32$ cells from five mice; two-tailed Student's $t$-test, $P = 0.002$). The fluorescence intensity of the 5-HT₆R-positive neurons (red) merged with GFP (green) was plotted using the same imaging conditions for every slice. **d**, **e** Western blots showing 5-HT₆R reduction after shRNA virus injection ($n = 4$ experiments/group; two-tailed Student's $t$-test, $P = 0.032$). **f–i** Knocking down 5-HT₆R in the GABAergic interneurons prevented EPA-induced enhancement of sIPSC frequency ($n = 8$ cells/group; two-tailed Student's $t$-test; **f**: Ctrl, $P = 0.021$ and shRNA, $P = 0.505$, **g**: Ctrl, $P = 0.789$ and shRNA, $P = 0.917$) and EPA-impaired LTP (**h**, **i**: $n = 7$–8 slices/group; two-tailed Student's $t$-test, $P = 0.001$). Scale bars: 0.5 mV, 5 ms. (**j–p**) The i.g. administration of EPA did not impair LTP (**j**, **k**: $n = 7$ slices/group; two-tailed Student's $t$-test; $P < 0.0001$) or learning and memory (MWM test: $n = 9$–10 mice/group; **l**: repeated measures two-way ANOVA, $F_{(1, 68)} = 43.369$, $P = 0.003$; **m**: two-tailed Student's $t$-test; $P = 0.029$; **n**: two-tailed Student's $t$-test; $P = 0.045$; **o**: two-tailed Student's $t$-test; $P = 0.002$. fear conditioning test in **p**: $n = 10$–11 mice/group; two-tailed Student's $t$-test; $P = 0.038$) after knockdown of 5-HT₆R in the CA1 GABAergic interneurons. Data show mean ± s.e.m. *$P < 0.05$, **$P < 0.01$, ***$P < 0.001$.

effect on basal synaptic transmission in terms of I–O curves or at baseline, PPF and sEPSCs. However, acute EPA application increased the frequency of sIPSCs and the amplitude of HFS-evoked IPSC. Meanwhile, we found that 10 μM BMI with no effect on LTP can prevent the EPA impairment of LTP. In addition, GABA_A receptor agonists can mimic the inhibitory effects of EPA on LTP induction as reported previously[55]. Therefore, it is possible that EPA-enhanced GABAergic neurotransmission may reduce the HFS-induced postsynaptic depolarization which is necessary for removing the Mg$^{2+}$ blockage of NMDARs[56], and thereby impair LTP induction. Immunofluorescent staining showing the localization of 5-HT₆R in the hippocampal GABAergic interneurons of GAD-GFP mice supported the hypothesis that 5-HT₆R is involved in the EPA-mediated regulation of GABAergic transmission. Indeed, the damaging effects of EPA on learning and memory and LTP can be prevented by a 5-HT₆R antagonist or genetic deletion of the receptor. Furthermore, selectively knocking down the receptor in GABAergic interneurons can block the impairment of synaptic plasticity and behaviour by EPA. Altogether, our study provides the first direct evidence that 5-HT₆R is expressed in the GABAergic interneurons of the hippocampus and that activating

this receptor can enhance GABAergic neurotransmission and suppress synaptic plasticity. We also observed an expression of the 5-TH₆R in pyramidal neurons. However, the mechanisms underlying EPA preferential modulation of GABAergic neurons remain unknown and need to further investigation. Interestingly, previous studies have demonstrated that, although their receptors are expressed both on GABAergic and glutamatergic neurons, but insulin or leptin only modulate GABA interneuron but not glutamatergic neuron[57,58].

In contrast to EPA in our study, supplementation with marine fish oil or combined EPA and DHA has not been reported to impair learning and memory[44]. Furthermore, when we compared the composition of various representative samples of these commonly consumed fish oils as well as EPA and DHA supplements, we found that they contained different EPA and DHA in different ratios[45,46], ranging from 1:1 to 1:2 in different studies. Therefore, we hypothesized that the ratio of EPA to DHA is critical for learning and memory. By administering EPA and DHA to mice at different ratios, we found that the combination of these fatty acids at a natural ratio can prevent EPA-induced impairments of learning and memory and LTP. Meanwhile, in screening the potential molecular targets underlying the effect of

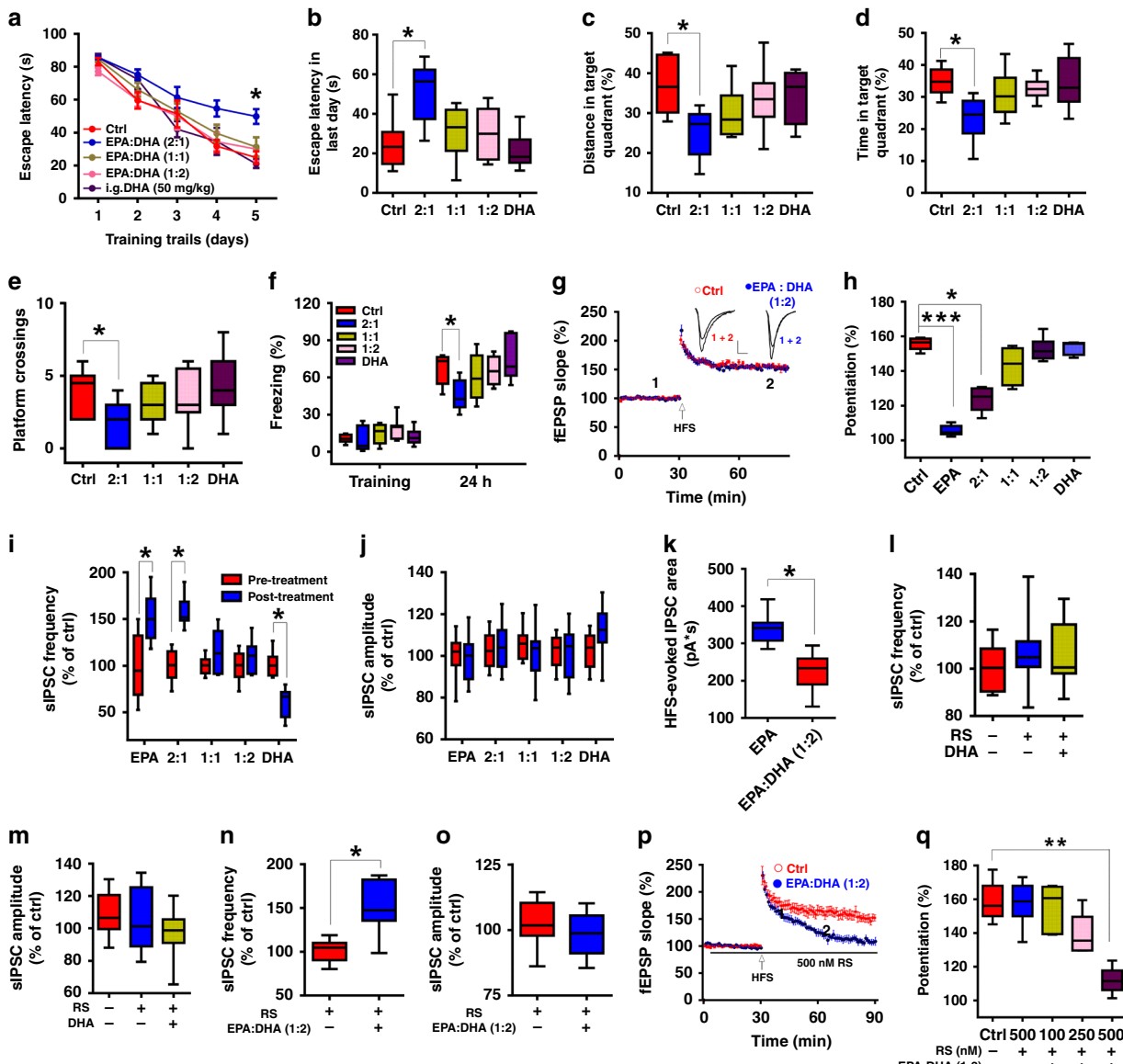

**Fig. 8 DHA prevents EPA-induced impairments via the 5-HT$_{2C}$R. a**, **b** Mean escape latencies across 5 consecutive days (**a**) or on the last day (**b**) during the MWM training (n = 10–11 mice/group; **a**: repeated measures two-way ANOVA, $F_{(4, 200)} = 33.354$, $P = 0.048$; **b**: one-way ANOVA, $F_{(4, 47)} = 33.456$, $P = 0.043$). **c**, **d** Percentage of distance (**c**) or time (**d**) spent in target quadrant during the probe trials (one-way ANOVA; **c**: $F_{(4, 49)} = 36.815$, $P = 0.036$; **d**: $F_{(4, 49)} = 41.46$, $P = 0.038$). **e** Number of platform crossings during the probe trials (one-way ANOVA; $F_{(4, 49)} = 9.681$, $P = 0.042$). **f** Freezing time in the contextual fear conditioning test (n = 10–11 mice/group; one-way ANOVA; $F_{(4, 49)} = 38.645$, $P = 0.031$). **g**, **h** HFS-induced LTP under different EPA/DHA ratio (n = 7–10 slices/group; one-way ANOVA; $F_{(5, 46)} = 22.132$, $P = 0.013$). Scale bars: 0.5 mV, 5 ms. **i**, **j** sIPSCs recorded under different EPA/DHA ratio (n = 9 cells; two-tailed Student's t-test; **i**, EPA: $P = 0.013$; 2:1: $P = 0.01$; 1:1: $P = 0.12$; 1:2: $P = 0.863$; DHA: $P = 0.022$. **j**, EPA: $P = 0.85$; 2:1: $P = 0.541$; 1:1: $P = 0.172$; 1:2: $P = 0.363$; DHA: $P = 0.473$). **k** DHA prevented the EPA enhancement of HFS-evoked IPSCs in hippocampal CA1 neurons (n = 10 cells from 4 mice; two-tailed Student's t-test; $P = 0.031$). **l**, **m** DHA was unable to inhibit sIPSCs in the presence of the 5-HT$_{2C}$R antagonist (n = 9 cells; one-way ANOVA, **l**: $F_{(2, 24)} = 11.78$, $P = 0.544$; **m**: $F_{(2, 24)} = 14.08$, $P = 0.753$). **g–m**: EPA: 50 μM, DHA: 100 μM. **n**, **o** EPA/DHA at the ratio of 1:2 increased the frequency of sIPSCs in the presence of the 5-HT$_{2C}$R antagonist (n = 9 cells; two-tailed Student's t-test; **n**, $P = 0.018$; **o**, $P = 0.245$). **p**, **q** EPA/DHA at the ratio of 1:2 suppressed LTP in the presence of the 5-HT$_{2C}$R antagonist (n = 7–9 slices/group; one-way ANOVA, $F_{(4, 39)} = 33.29$, $P = 0.021$). Data show mean ± s.e.m.. *$P < 0.05$, **$P < 0.01$, ***$P < 0.001$.

DHA, we found through a radio-ligand receptor binding assay that 5-HT$_{2C}$R is physically suited to be a specific binding site of DHA. As the cAMP pathway is among the reported intracellular pathways that could contribute to 5-HT$_{2C}$R-dependent signaling including cAMP[59], it is reasonable to assume that DHA may exert its function through suppression of the cAMP pathway. Consistent with previous reports[60], we also found a DHA inhibition of GABAergic transmission, which can be blocked by the 5-HT$_{2c}$R antagonist. Furthermore, the antagonist could block the

restorative effect of DHA on EPA-impaired sIPSCs and LTP. These results suggest that DHA may prevent the EPA-induced impairment of learning and memory and synaptic plasticity by antagonizing enhanced GABAergic transmission. It is accepted that blockade of GABAergic neurotransmission facilitates LTP induction[61,62], whereas in our study, acute DHA treatment did not affect LTP but partially inhibited GABAergic neurotransmission. This discrepancy may be due to the difference in the extent to which GABAergic neurotransmission was

suppressed. In support of this notion, our results in Fig. 6m showed that complete, but not partial, inhibition of GABAergic neurotransmission enhanced LTP.

Previous studies also investigated the effects of omega-3 fatty acids including EPA and DHA on impaired synaptic plasticity in aging and pathological conditions. For example, chronic administration of omega-3 fatty acids reversed age-related[63,64], and amyloid-beta-[65], lipopolysaccharide-[66] and irradiation-induced[67] deficits in LTP in the dentate gyrus. The underlying mechanisms probably involve antioxidant and anti-inflammatory activity or alterations in the membrane lipid composition. These investigations help clarify the mechanistic role of omega-3 fatty acids in the modulation of synaptic plasticity.

Taken together, our studies demonstrate that acute EPA administration is detrimental to learning and memory and synaptic plasticity and reveal its underlying molecular mechanism in mice. Our findings also suggest the exciting possibility that, by combined administration with DHA at a natural ratio or blocking 5-HT$_6$R, one may prevent the detrimental effects of EPA while gaining its beneficial effects.

## Methods

**Animals**. Adult male C57BL/6J mice (purchased from the Guangzhou Southern Medical University Animal Center) were housed in standard laboratory cages (4–5 per cage), with a 12-h light/dark cycle (lights on at 8:00 A.M.), in a temperature-controlled room (21–25 °C). Mice were provided free access to food and water. All experiments were conducted in accordance with the Regulations for the Administration of Affairs Concerning Experimental Animals (China), and were approved by the Southern Medical University Animal Ethics Committee. The behavioral tests were performed by experimenters who were blinded to the experimental group.

The 5-HT$_6$R KO (5-HT$_6$R$^{-/-}$) mice, which were purchased from the Mutant Mouse Resource & Research Center, were generated by crossing germline-heterozygous-null mutants. The offspring were genotyped by PCR using DNA isolated from the tail tissue, and WT (5′-AGAGCCCGGCCCTGTCAAC -3′; 3′-AG GGCACGCGGGGCTGTCAT-5′) and mutant allele-specific primers (neospecific primer 5′-GCACGCGCATCGCCTTCTATC-3′; 3′-CCCGTCACAGAGAAGCATG CCAGC-5′). The PCR products were visualized using ethidium bromide staining.

The fat-1 transgenic mice were donated by Professor Xiaochun Bai. The male Fat-1-positive and Fat-1-negative mice were genotyped by PCR using DNA isolated from the tail tissue (5′-GCCGTCGCAGAAGCCAAAC-3′; 5′-GGACCTG GTGAAGAGCATCCG-3′). The Fat-1 gene of C. elegans encodes an n-3 fatty-acid desaturase enzyme which converts n-6 to n-3 fatty acids[32]. Therefore, the fatty-acid profiles of fat-1 transgenic mice exhibit a high n-3/n-6 level, if maintained on a diet that is high in n-6 but deficient in n-3 fatty acids. However, in our study, the diet is standard diet, just like the control diet from previous studies[32,68–71].

**Drugs**. Eicosapentaenoic acid (EPA; Sigma-Aldrich, USA) and docosahexaenoic acid (DHA, Sigma-Aldrich, USA) were dissolved in dimethyl sulfoxide (DMSO), and the concentration of DMSO (Sigma) used for all solutions was less than 0.1%. The EPA concentration used in our study was derived from the FDA-recommended human dosage of 3 g per day of EPA plus DHA to reduce the risk of cardiovascular disease (CVD)[23]. Therefore, an adult human (75 kg) would take 40 mg/d/kg of EPA plus DHA (for EPA only, the dosage would be 20 mg/d/kg). According to the Meeh-Rubner formula[72], the conversion coefficient between human and mouse doses is 1:8, accordingly, we converted the human EPA dose of 20 mg/d/kg to an animal equivalent dose for mice, that is, 160 mg/d/kg. SB-399885 (SB, Sigma-Aldrich, USA) and EMD-368088 (EMD, Sigma-Aldrich, USA) were dissolved in distilled water. All other chemicals were purchased from Sigma Aldrich.

**Behavioral assays**. All behavioral tests were conducted during the dark period of the circadian cycle (1:00–5:00 P.M.).

**Morris water maze test**. The MWM test was conducted as reported previously[24,73]. The experimentation room contained several permanent extra-maze cues, and the tank was divided into four quadrants arbitrarily denoted NE, NW, SE, and SW. Swimming behavior was captured using a camera (Panasonic WVBP334, Suzhou, China), and the video was analyzed using EthoVision 7.0 (Noldus Information Technology, Leesburg, VA, USA). During acquisition, each animal completed four trials per day for four or five consecutive days. The escape platform, 10 cm in diameter, was placed in the center of the first quadrant, submerged 1.0 cm beneath the water surface. The mouse was gently placed in the water, between quadrants, facing the wall of the pool. The drop location varied for each trial, and the mouse was allowed 90 s to locate the submerged platform. If the

animal failed to identify the platform within 90 s, it was guided gently onto the platform, and allowed to remain there for 30 s. Escape latency was measured as the time taken for the animal to locate the hidden platform in the target quadrant. Each animal was subjected to training trials for four or five consecutive days. The time required to locate the platform (latency) and velocity were measured. For the probe test on day 5, the platform was removed, and the total distance traveled, time spent in the target quadrant, and the numbers of platform crossings were monitored for 60 s. Two hours later, the platform was replaced, and the ability of the animals to locate it within 60 s was assessed. Data of training process were analyzed using repeated measures two-way ANOVA and data of test process were analyzed using at wo-tailed Student's t-test or one-way ANOVA and the Bonferroni test.

**Contextual fear conditioning test**. The contextual fear conditioning test was conducted as reported previously[74–76]. Mice were first habituated to the behavioral room, and were then allowed to freely explore the apparatus (MED-VFC-NIR-M; Med Associates) for 3 min. During training, mice were placed in a conditioning chamber A, and exposed to tone-foot-shock pairings (tone, 30 s, 80 dB; foot shock, 1 s, 0.4 mA), with an interval of 80 s, 24 h after training. Mice were returned to the chamber A to evaluate contextual fear learning. Two hours later, cued fear conditioning was performed. Each mouse was placed into novel chamber B, monitored for 3 min. Freezing during training and testing was scored using Med Associates Video-Tracking and Scoring software. Data were analyzed using a two-tailed Student's t-test or one-way ANOVA and the Bonferroni test.

**Novel object recognition test**. The NOR test was conducted as previously described[26]. Mice were allowed to explore two identical objects placed in a 30 × 50-cm arena (10-min exploration) on day 1, returned to their home cage immediately after training, and were tested 24 h later, when one of the two objects had been replaced with a new one (5 min exploration). Discrimination indexes were calculated as $(t_{novel} - t_{familiar})/(t_{novel} + t_{familiar})$. To avoid discrimination of the objects based on odor, both the arena and the objects were thoroughly cleaned with 70% ethanol before and after each trial. Data were analyzed using a two-tailed Student's t-test or one-way ANOVA and the Bonferroni test.

**Open field test**. The open field test was performed in a rectangular chamber (60 × 60 × 40 cm) made of gray polyvinyl chloride, the central area of which was illuminated by 25 W halogen bulbs (200 cm above the field). Mice were gently placed into the testing chamber for a 5-min recording period, which was monitored using an automated video-tracking system. Images of the paths traveled in the 5 min were automatically calculated using the DigBehv animal behavior analysis program. Data were analyzed using a two-tailed Student's t-test or one-way ANOVA and the Bonferroni test.

**Pain threshold test**. The pain threshold test was conducted as previously described[75]. A mouse was placed into chamber A and received 11 repeated scrambled shocks with various intensities (0.10, 0.15, 0.20, 0.25, 0.30, 0.35, 0.40, 0.45, 0.50, 0.55, and 0.60 mA). The shock lasted 1 s and the intershock intervals were at least 2 min. Two experimenters without prior knowledge of shock intensities or genotypes scored the flinching, vocalization or jumping response. A flinching event was defined as when the mouse curled up their feet, vocalization as when the mouse made an audible squeak and jumping as when the mouse propelled itself off the floor. Data were analyzed using repeated measures two-way ANOVA.

**Olfactory habituation/dishabituation test**. The olfactory habituation/dishabituation test was performed in a hamber consisting of an open-top plastic box (12 × 12 × 26 cm) with a recessed odor port on one side to provide odorant delivery. Each animal was tested once per day within a single session lasting from 15 to 70 min (mean, 45 min; 2–3 sessions per animal).Each of the 3 odorants was presented 3 consecutive times for a duration of 120 s, followed by a 1- to 2-min intertrial interval (ITI). Each odorant was presented three times to ensure robust habituation to the test stimulus. To establish a baseline for sniffing behavior and to control for airflow changes in the chamber, a "blank" odorant was presented in the same manner at the start of each session. Odorants used included almond, banana and the excreta of a different animal (e.g. C57Bl/6J or B6.129S6 mice). All odorants were presented at 0.5% saturated vapor. The testing chamber was cleaned with 70% ethanol between each mouse. The time spent sniffing the tip during the 2-mintrial was recorded. Data were analyzed using repeated measures two-way ANOVA.

**T-maze test**. The T-maze test was conducted in a T shaped elevated maze (30 × 10 cm start arm and two 30 × 10 cm goal arms, with stripes or circles on the walls of the goal arms). During training, both of the two arms were open, mice were placed in the start arm facing away from the choice point, and allowed to freely explore the maze for 20 min. Once the mice entered the arm, we closed the choiced door. The mice were return to a new cage 1 min later. The retention test was performed 5 min after training. During this test both goal arms were open, mice were placed at the end of the start arm facing away from the choice point and allowed to freely

explore for 5 min. The percent of time spent in each arm and the total exploration (in meters) were measured using the EthoVision tracking software (Noldus). New arm preference was calculated by dividing the percentage of time spent in the new arm (that was closed during training) by the percentage of time spent in both goal arms (new and old)[77]. Data were analyzed using a two-tailed Student's t-test.

### Electrophysiology analysis

*Slice preparation.* Coronal hippocampal slices were prepared as described previously[75]. In brief, mice were decapitated, and transverse hippocampal slices (300-μm-thick) were prepared using a Vibroslice (VT 1200S; Leica) in ice-cold artificial cerebrospinal fluid (ACSF).The slice-cutting solution contained (in mM): 220 sucrose, 2.5 KCl, 1.3 $CaCl_2$, 2.5 $MgSO_4$, 1 $NaH_2PO_4$, 26 $NaHCO_3$, and 10 glucose, whereas the recording ACSF contained (in mM): 126 NaCl, 26 $NaHCO_3$, 3.0 KCl, 1.2 $NaH_2PO_4$, 2.0 $CaCl_2$, 1.0 $MgSO_4$, and 10 glucose. After cutting, the hippocampal slices were left for "recovery" in the chamber for 30 min at 34 °C, and then at room temperature (25 ± 1 °C) for an additional 1 h. All solutions were saturated 95% $O_2$/5% $CO_2$ (vol/vol).

*Electrophysiological recordings.* Electrophysiological Recordings were performed as described previously[75,78]. Slices were placed in the recording chamber, which was superfused (3 mL/min) with ACSF at 32–34 °C. The fEPSPs were evoked in the CA1 stratum radiatum by stimulating the SC with a two concentric bipolar stimulating electrodes (FHC), and recorded in current-clamp mode, using the Axon MultiClamp 700B (Molecular Devices) amplifier with ACSF-filled glass pipettes (3–7 MΩ). The test stimuli consisted of monophasic 100-μs pulses of constant currents (with intensity adjusted to produce 25% of the maximum response), at a frequency of 0.033 Hz. The strength of synaptic transmission was determined by measuring the initial (20–70% rising phase) slope of fEPSPs. All drugs were dissolved in ACSF, and applied by switching the perfusion from control ACSF to drug-containing ACSF. In each recording, baseline synaptic transmission was monitored for 10 min before drug administration, and the average of fEPSP slopes during the 20 min prior to the induction of LTP was considered as the baseline, and all values were normalized to this baseline. The LTP was induced by one train of 100-Hz stimuli, each having 50 pulses, separated by 10 s, with the same intensity as that of the test stimulus; LTP was also induced by four trains of TBS, each of 4 pulses, at 100 Hz, with 200-ms interval. The extent of LTP was determined at an average of 30–60 min after tetanic stimulation. The PPF was examined by applying pairs of stimuli at varying inter-pulse intervals (20–200 ms). The slope of the response to the second pulse (P2) was averaged over 5–10 trials, and divided by the average slope or amplitude of the response to the first pulse (P1) to obtain a ratio (P2/P1). Data of LTP induction were analyzed using a two-tailed Student's t-test or one-way ANOVA and the Bonferroni test. Data of I–O curve and PPR were analyzed using repeated measures two-way ANOVA.

### Whole-cell patch-clamp recordings

Whole-cell patch-clamp recordings from the CA1 neurons were visualized with infrared optics, using an upright microscope equipped with a 40× water-immersion lens (BX51WI; Olympus) and infrared-sensitive CCD camera. For sEPSC recording, neurons were held at −70 mV in the presence of 20 μM BMI, with the pipette solution containing (in mM): 125 cesium methanesulfonate, 5 CsCl, 10 Hepes, 0.2 EGTA, 1 MgCl2, 4 Mg-ATP, 0.3 Na-GTP, 10 phosphocreatine and 5 QX314 (pH 7.40, 285 mOsm). For sIPSCs recording, neurons were held at −70 mV and pipettes were filled with an intracellular solution containing (in mM): 110 $Cs_2SO_4$, 0.5 $CaCl_2$, 2 $MgCl_2$, 5 EGTA, 5 HEPES, 5 TEA, 5 Mg-ATP (PH 7.3, 285 mOsm). For recording NMDAR current, we blocked AMPAR using 20 μM CNQX. For HFS-evoked IPSC recording, the holding potentials were 0 mV, pipettes (input resistance:3–7 MΩ) were filled with an intracellular solution containing (in mM): 115 cesium methanesulphonate, 20 CsCl, 10 HEPES, 2.5 $MgCl_2$, 10 sodium phosphocreatine, 5 QX-314, 4 $Na_2$-ATP, 0.4 $Na_3$GTP, and 0.6 EGTA (pH 7.3, 285 mOsm). Synaptic responses were evoked by a stimulating electrode HFS (one train of 100-Hz) that placed in the CA3 region. Data were analyzed using a two-tailed Student's t-test. Data of NMDA currents were analyzed using repeated measures two-way ANOVA.

In all experiments, series resistance was controlled below 20 MΩ and not compensated. Cells would be rejected if membrane potentials were more positive than −0 mV; or if series resistance fluctuated more than 20% of initial values. All recordings were done at 32–34 °C. Data were filtered at 1 kHz and sampled at 10 kHz.

### Fatty acid composition of mouse tissues

Fatty acid profiles of mouse brain tissues were analyzed using gas chromatography as described previously[79]. Briefly, tissues were homogenized by grinding in liquid nitrogen, and subjected to fatty acid methylation by mixing with 1 ml of hexane and 1 ml of 14% BF3/MeOH reagent at 100 °C for 1 h. Fatty acid methyl esters were extracted in the hexane phase, and then the fatty acid profiles were analyzed using a fully automated HP6890 gas chromatography system equipped with a flame-ionization detector (Agilent Technologies, Palo Alto, CA, USA). The fatty acid peaks were identified by comparing relative retention times with commercial mixed standards (Nu-Chek Prep, Elysian, MN, USA), and the area and its percentage for each peak were

analyzed using GC Chemstation software. Data were analyzed using a two-tailed Student's t-test.

### Real-time quantitative PCR

The quantitative real-time PCR was done on a Stratagene Mx3000P thermal cycler using Universal qRT-PCR master mix for the indicated genes (Takara). The primers were designed and synthesized as follows:
FABP7 sense: 5′-ATGGAGACAAGCTCATTCATGTG-3′, antisense: 5′-TGC CTTTTCATAACAGCGAACA-3′. fat1 sense: 5′-CGCCACGATTACTCTCA ATAA-3′, antisense: 5′-TTATTGAGAGTAATCGTGGCG-3′Gapdh sense 5′ -GGC ACAGTCAAGGCTGAGAATG-3′, antisense 5′-ATGGTGGTGAAGACGCCA GTA-3′. Expression of target genes was normalized against the expression of Gapdh as an endogenous control gene. Data were derived from cells from three independent cultures from at least 3 litters. Data were analyzed using a two-tailed Student's t-test.

### Radio-ligand receptor binding assays

Binding affinity was determined by competition with [3H]paroxetine (PerkinElmer Life Sciences, LesUlis, France)[80]. Freshly prepared membranes of the rat frontal cortex were homogenized using a Polytron homogenizer, and then centrifuged twice at $20,000 \times g$. The pellet was resuspended each time in an incubation buffer. Membranes were incubated in triplicate with 2 nM [3H]paroxetine and competing ligand in a final volume of 0.4 ml for 2 hat 25 °C. The incubation buffer contained 50 nM Tris-HCl (pH 7.4), 120 nM NaCl, and 5 mM KCl. Nonspecific binding was defined by 10 mM citalopram. At the end of the incubation period, membranes were filtered through Whatman (Packard, Meriden, CT, USA) GF/B filters pretreated with 0.1% polyethylenimine. Radioactivity retained on the filters was determined using scintillation counting. Binding isotherms were analyzed using nonlinear regression using GraphPad Prism software (GraphPad Software Inc., San Diego, CA, USA) to determine $IC_{50}$ values. These were converted to inhibition constants ($K_i$) by using the Cheng-Prusoff equation: $K_i = IC_{50}/[(L/K_D) − 1]$, where L is the concentration of 3H-labeled ligand and $K_D$ is its dissociation constant, determined in saturation binding experiments. The $K_D$ values of rat SERTs were 0.13 nM for [3H]paroxetine.

### cAMP accumulation assay

The 293 T cells stably expressing human and mouse 5-$HT_6R$ were transfected in 96-well plates, at a density of 50,000 cells per well[81]. Various concentrations of EPA and 5-$HT_6R$ agonists and antagonists were added to the cells to stimulate the receptor 24 h after transfection, and 0.5 h later the cAMP level was monitored using the Cyclic AMP XP™ Assay Kit (Cell Signaling Technology)[75].Data were analyzed using atwo-tailed Student's t-test or one-way ANOVA and the Bonferroni test.

### Western blots

Western blotting was performed as described previously[81]. Briefly, protein concentration was measured using the BCA Protein Assay Kit (Thermo). Samples were mixed with 2x sodium dodecyl sulfate (SDS) loading buffer, boiled for 10 min, and loaded onto 10% or 4–20% gradient polyacrylamide-SDS gel. Proteins were then transferred to a PVDF membrane (Millipore, Billerica, MA, USA) for 2 h at 350 mA, and membranes were incubated in Odyssey Blocking Buffer (LiCor) for 2 h at room temperature. After overnight incubation with primary antibodies at 4 °C, the blots were washed three times in TBS containing 0.1% Tween-20 for 15 min, and then incubated with peroxidase- or IRDye-conjugated secondary antibody for 1 h in TBS, 0.1% Tween-20 at room temperature. Immunoreactivity was detected by chemiluminescence using ECL reagent and LiCor imaging system. Data were analyzed using a two-tailed Student's t-test.

### Virus generation and stereotaxic microinjection

Stereotaxic Microinjection was performed as described previously[82]. The recombinant adeno-associated viral (AAV) vectors were generated by Shanghai Sunbio Medical Biotechnology (Shanghai, China) and were ligated into an AAV5 vector expressing EGFP with viral titers of $2 \times 1012$ particles/ml. The micropipette was brought to the correct x and y coordinates and lower to the desired z coordinate of the injection site. A 33-gauge needle fitted to a Hamilton syringe was lowered to the hippocampal CA1 region (AP, − 2.0 mm; ML, ± 1.6 mm; DV, –1.5 mm), and 0.25 μl (0.1 μl/min) of the virus was delivered over 3 min. The needle was withdrawn 10 min after the end of injection. Mice were used 3 weeks after AAV injection.

### Immunofluorescence

Mice were anesthetized using pentobarbital sodium (50 mg/kg), and infused with saline, followed by 4% formaldehyde, from the base of the left ventricle. Brains were cut into slices of 40 μm thickness using a freezing microtome (Leica). Slices were washed with phosphate-buffered saline, and treated with 1% Triton-100, followed by goat serum, and overnight incubation with the primary antibody (rabbit anti-5-HT6R [1:1000; Abcam, Cambridge, MA, USA] and mouse anti-GAD67 [1:500; MAB5406; Millipore]) at 4 °C. Slices were then incubated with the corresponding fluorophore-conjugated secondary antibody (Alexa Fluor 488 [1:500; A11034; Invitrogen], Alexa Fluor 594 [1:500; A11005; Invitrogen]) at room temperature for 1 h. The slices were then mounted using Fluoroshield mounting medium with 4′,6- diamidino-2-phenylindole (ab104139; Abcam). Images with fluorescence were captured by fluorescent microscopy (Nikon).

**Microdialysis**. Microdialysis was performed to determine 5-HT levels in adult male C57BL/6J mice or fat-1 mice. Each mouse was then deeply anesthetized, and mounted on the stereotaxic frame (Stoelting). A guide cannula (CMA/7, CMA/ Microdialysis) was implanted into the hippocampal CA1 region (AP = −2.0 mm; ML = 1.6 mm; DV = −1.5 mm). A microdialysis probe (CMA/7, membrane length: 1 mm, molecular weight cut-off: 6,000 Da, outer diameter: 0.24 mm) was inserted through the guide cannula, and connected to the syringe pump (CMA 402); 24 h later, the ACSF was continuously perfused through the microdialysis probe at a constant flow rate of 1 μl/min, and sampling was performed 1 h following the insertion of the probe. Samples (30 μl each) were automatically collected from each mouse using the CMA 142 microfraction collector, every 30 min, till the end of the experiment. To decrease the rate of 5-HT oxidation, each sample collection tube was pretreated with antioxidative agents (8 μl), containing 100 mM acetic acid, 0.27 mM disodium EDTA, and 12.5 μM ascorbic acid (pH 3.2)[83], and the interstitial fluid 5-HT levels were measured immediately using high performance liquid chromatography with electrochemical detection. Data were analyzed using a two-tailed Student's t-test or repeated measures one-way ANOVA.

**Statistical analyses**. The data were expressed as mean ± standard error of the mean. The statistical analyses were performed using SPSS 13.0 software. Differences between the mean values were evaluated using one-way analysis of variance followed by the least significant difference test for post hoc comparisons when equal variances were assumed. Independent samples t-tests were used to compare differences between any two given groups throughout the study, unless otherwise specified. The significance level for all the tests was set at $P < 0.05$.

**Reporting summary**. Further information on research design is available in the Nature Research Reporting Summary linked to this article.

## Data availability
The data that support the findings of this study are available from the corresponding author on reasonable request. Source data are provided with this paper.

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

## Acknowledgements

This work was supported by grants from the National Natural Science Foundation of China (31830033, 81671356, 31600864), the Program for Changjiang Scholars and Innovative Research Team in University (IRT_16R37), Key-Area Research and Development Program of Guangdong Province (2018B030334001), the Guangzhou Science and Technology Project (201707020027), the China Postdoctoral Science Foundation (2018M633072), and Guangdong-Hong Kong-Macao Greater Bay Area Center for Brain Science and Brain-Inspired Intelligence Fund (2019020).

## Author contributions

T.G. and J.L. designed the study. J.L., Q.Y., and Z.L. performed the behavior tests and analysis. J.L. performed the electrophysiology recordings and Q.Y. conducted in vitro patch-clamp experiments. Q.W. and Y.K. conducted the ELISA measurements and microdialysis analysis. N.H. performed the cell culture and immunofluorescence. Y.W. X. B., and Y.D. performed the GC experiments and analysis. Z.J. and J.X. performed the Radio-ligand Receptor Binding Assay and IC50 analysis. S.L., X.Z., and X.L. performed the western blotting. T.G. conceived the project and wrote the paper with the assistance of J.L. and J.Y.

## Competing interests

The authors declare no competing interests.
