## [Peer Review File · Nature Communications]

Reviewers' Comments:

Reviewer #1:

Remarks to the Author:

This is a wonderfully detailed, mechanistic and clinically important paper. I am disappointed that other reviewers did not see it like I did. This should have been published without delay more than a year ago... Fantastic paper!

Reviewer #2:

Remarks to the Author:

The manuscript of Liu et al reports that the fatty acid EPA has detrimental effects of synaptic plasticity and different forms of hippocampus-dependent learning through actions on GABAergic transmission that are mediated by the serotonin 5-HT₆ receptor. The huge body of results presented is novel, very impressive and even greater in the revised manuscript. Among the additions and edits in the revised report, it is important that they further demonstrate that the effects of EPA depend upon increases in GABAergic inhibition (in particular they provide evidence that inhibiting GABAergic function can prevent deleterious effects of EPA on LTP without influencing the LTP itself). Among the many electrophysiological results presented the finding that the serotonin receptor antagonist acutely blocked the effect of EPA on LTP is impressive and, to me, rather convincing as to mechanism of action....for LTP at least.

In a second set of new experiments they show that the ratios of applied EPA to DHA is critical for effects of treatments on synaptic plasticity. The latter is important as it places these studies more reasonably within the context of prior work that had failed to find deleterious effects of fish oil on cognitive function.

In all, the manuscript is much improved upon revision. I do believe that further revision, or to be more precise, editing, is needed. As listed below:

Abstract:

The abstract is very confusing. Having just read the responses to the review I know what they are intending to say but the current wording does not convey the meaning they intend. The comment that the impact of EPA on "the molecular targets" is confusing. What molecular targets are they referring to? I suspect they wanted to say that the molecular targets of EPA are not known - but that is not what the sentence says. Moreover, when they refer 'a ratio', it is not clear that they mean the ratio of EPA to DHA. The abstract needs to be rewritten.

Introduction:

Some of the problematic sentences in the abstract are repeated in the introduction. In particular on page 3, line 7, I believe they meant to say "that contain high amounts of...". And at the end of the same paragraph I believe their meaning is better conveyed if the sentence says "...cognitive function or the molecular target of omega-3 fatty acids in brain."

In the next paragraph...I believe the wording is in error in that they cannot study learning and memory 'ex vivo'. Also they need to define "5-HT_{6R}".

For LTP effects they report that the expression of LTP is impaired after peripheral EPA treatment in vivo, as evaluated in slices. The timing is not given in the text, the brief methods, or the captions and as it is quite important (how many minutes or hours later was LTP assessed) – how can this be the

case if EPA had been perfused out of the slices - this information should be immediately available in the text.

In the Results, page 8, 7 lines down, the word 'molecule' is not correct...this would read better if it stated something like "...a downstream 'element within' (or 'target of') the".

On the same page (8) they should indicate if treatment with SB was given 30 min before training or just testing (this is a critical point pertinent to whether treatment influenced learning or recall). Also the caption on this should be specific and not say that it 'blocked the impairment in learning and memory'....did it block learning (acquisition)? Or memory?

The first added sentence on page 10 has language problems and would read better as: "EPA also increased the feedforward IPSCs evoked by the HFS used to induce LTP (Fig. 6e,f), suggesting that EPA impaired LTP by enhancing GABAergic transmission".

On page 10, BMI is not defined.

The added block of text on page 11 needs language editing. These new results are impressive and extremely important in addressing comments of the third reviewer but editing of the paragraph is needed. For example, I believe the third sentence down should read:

"... in various representative experiments showed that they contained ratios of EPA and DHA in the 1:1~1:2 range^{42, 43}, suggestion the ratio of EPA to DHA is critical for effects on learning and memory and other physiological process." This is just one example.

On page 15, in the first inserted block of text, there are edits needed although the meaning is clear and comments are appropriate.

In all, the revisions have made the manuscript much stronger and addressed major concerns of the second the third reviewers.

Reviewer #3:

Remarks to the Author:

This is an interesting paper examining acute doses of EPA on behaviour. The authors report acute EPA administration impairs memory and have done many experiments that this is dependant on 5-HT_{2c}R. While overall this is an interesting paper, the brain lipid chemistry data seems to be a odds with the literature and EPA has been given to humans at high doses (5 g per day for years) without reports of memory impairment suggesting something unique about the acute model being studied.

The lengthy background on CVD is a bit distracting and could be shortened. It isn't clear why the authors should focus on CVD when their work is brain-specific. There is a relatively large literature on EPA and major depression that is not commented on. See meta-analyses by Su and others suggesting benefits of EPA.

One of the major concerns about this study is it potential translation to humans. Numerous studies have been conducted with EPA and no detriments have been reported (in terms of cognitive function) if anything observational data suggests a protective effect. It seems most likely that the authors effects are model specific. Is this a result of acute dosing? Are the results seen with chronic intake? Another major concern with this report is that lack of consideration of the interdependency of EPA and DHA. EPA is readily converted to DHA (Metherel AJCN 2020). How is it possible that EPA increased EPA but not DHA in the brain regions? Is this due to the acute nature of the studies? Also the large differences in DHA concentrations across brain regions (PFC, striatum and Hip) are not consistent with

the literature and because they occur for all fatty acids suggests an artifact.

How do the authors explain that EPA supplementation increased EPA selectively in the Hip (Figure 1h). Similarly speaking that fat-1 model is well known to increase brain DHA (including in the Hip – Hopperton BBI 2018; Orr JNC 201; Delpech Neuropsychopharmacology 2015). However, they do not see an increase in DHA figure S5. How is this possible? What was the background diet of these animals?

A point-by-point reply to the reviewers follows:

Reviewer #1: This is a wonderfully detailed, mechanistic and clinically important paper. I am disappointed that other reviewers did not see it like I did. This should have been published without delay more than a year ago... Fantastic paper!

==> We thank the reviewer for the careful reading and the positive comments: “This is a wonderfully detailed, mechanistic and clinically important paper” and “Fantastic paper”.

No comments.

Reviewer #2: The manuscript of Liu et al reports that the fatty acid EPA has detrimental effects of synaptic plasticity and different forms of hippocampus-dependent learning through actions on GABAergic transmission that are mediated by the serotonin 5-HT₆ receptor. The huge body of results presented is novel, very impressive and even greater in the revised manuscript. Among the additions and edits in the revised report, it is important that they further demonstrate that the effects of EPA depend upon increases in GABAergic inhibition (in particular they provide evidence that inhibiting GABAergic function can prevent deleterious effects of EPA on LTP without influencing the LTP itself). Among the many electrophysiological results presented the finding that the serotonin receptor antagonist acutely blocked the effect of EPA on LTP is impressive and, to me, rather convincing as to mechanism of action....for LTP at least.

In a second set of new experiments they show that the ratios of applied EPA to DHA is critical for effects of treatments on synaptic plasticity. The latter is important as it places these studies more reasonably within the context of prior work that had failed to find deleterious effects of fish oil on cognitive function.

In all, the manuscript is much improved upon revision. I do believe that further revision, or to be more precise, editing, is needed.

==> We thank the reviewer for the careful reading and the positive comments that “The huge body of results presented is novel, very impressive and even greater in the revised manuscriptAmong the many electrophysiological results presented the finding that the serotonin receptor antagonist acutely blocked the effect of EPA on LTP is impressive and, to me, rather convincing as to mechanism of action....for LTP at least” and “...The latter is important as it places these studies more reasonably within the context of prior work that had failed to find deleterious effects of fish oil on cognitive function” and “In all, the manuscript is much improved upon revision”. We also thank the reviewer for the constructive suggestions that have significantly improved the manuscript.

Major comments:

(1) The abstract is very confusing. Having just read the responses to the review I know what they are intending to say but the current wording does not convey the meaning they intend. The comment that the impact of EPA on "the molecular targets" is confusing. What molecular targets are they referring to? I suspect they wanted to say that the molecular targets of EPA are not known - but that is not what the sentence says. Moreover, when they refer 'a ratio', it is not clear that they mean the ratio of EPA to DHA. The abstract needs to be rewritten.

==> We thank the reviewer for the thoughtful comments and suggestions and we are sorry to make you confused. We have rewritten the abstract in our revised manuscript and also double-checked all the manuscript to avoid this kind of mistake.

(2) Some of the problematic sentences in the abstract are repeated in the introduction. In particular on page 3, line 7, I believe they meant to say "that contain high amounts of...". And at the end of the same paragraph I believe their meaning is better conveyed if the sentence says "...cognitive function or the molecular target of omega-3 fatty acids in brain."

==> We thank the reviewer for the thoughtful comments and suggestions and we are sorry to make you confused. We have rewritten the problematic sentences in our revised manuscript and also double-checked all the manuscript to avoid this kind of mistake.

(3) In the next paragraph...I believe the wording is in error in that they cannot study learning and memory 'ex vivo'. Also they need to define "5-HT₆R".

==> We thank the reviewer for the careful reading and constructive suggestions and we are sorry to make you confused. We have rewritten the problematic sentences and defined "5-HT₆R" in our revised manuscript and also double-checked all the manuscript to avoid this kind of mistake.

(4) For LTP effects they report that the expression of LTP is impaired after peripheral EPA treatment in vivo, as evaluated in slices. The timing is not given in the text, the brief methods, or the captions and as it is quite important (how many minutes or hours later was LTP assessed) – how can this be the case if EPA had been perfused out of the slices - this information should be immediately available in the text.

==> We thank the reviewer for the careful reading and constructive suggestions and we are sorry to make you confused. We have added the timing (We administered a series of concentrations of EPA to the mice. One hour later we sacrificed the mice and obtained the hippocampal slices immediately. After incubated in ACSF for another one hour, we recorded LTP and found that 50, 75 and 150 mg/kg EPA significantly suppressed high-frequency stimulation (HFS)-induced LTP in adult mice) in the text and the captions of Figure 1 (i, j) in our revised manuscript. As the reviewer pointed out, it is possible that EPA had been perfused out of the slices in this condition. However, in exploring the modulation of EPA on GABAergic

transmission, we found that perfusion of the slices with 50 μ M EPA led to an increase in the frequency of spontaneous inhibitory postsynaptic currents (sIPSCs) (Fig. 6b-d, and also see below) and this enhanced effect was not disappeared after one hour of washout with ACSF (see below, the later data was not showed in the manuscript). These results suggested that, once binding to the 5-HT₆R, EPA may activate the downstream target of the 5-HT₆R/adenylyl cyclase signaling pathway and the activated signaling can last for some time after removing EPA. Therefore, we recorded the suppressed LTP in the slices after one hour of perfusion.

Effects of EPA on GABAergic transmission. (A) Representative sIPSCs recorded from a CA1 pyramidal neuron before, during and washout of the application of 50 μ M EPA. Scale bars: 20 pA, 2 s. **(B, C)** Average sIPSC frequency (Hz) in **(B)** and amplitude in **(C)** (n=9 cells from 5 mice; one-way ANOVA; B: $F_{(2, 24)}=19.245$, $P=0.042$; C: $F_{(2, 24)}=11.242$, $P=0.468$).

(5) In the Results, page 8, 7 lines down, the word 'molecule' is not correct...this would read better if it stated something like "...a downstream 'element within' (or 'target of') the".

==> We thank the reviewer for the thoughtful comments and constructive suggestions. We are sorry to make this mistake. The "molecule" was changed to "target" in the revised manuscript.

(6) On the same page (8) they should indicate if treatment with SB was given 30 min before training or just testing (this is a critical point pertinent to whether treatment influenced learning or recall). Also the caption on this should be specific and not say that it 'blocked the impairment in learning and memory'....did it block learning (acquisition)? Or memory ?

==> Good suggestions. In fact, SB was given before both training on the first day and testing on the second day. So in the caption we say "blocked the impairment in learning and memory". We added the timing in the revised text.

(7) The first added sentence on page 10 has language problems and would read better as: "EPA also increased the feedforward IPSCs evoked by the HFS used to induce LTP (Fig. 6e,f), suggesting that EPA impaired LTP by enhancing GABAergic transmission".

==> Good suggestions. We have rewritten the problematic sentences in the revised manuscript as you suggested.

(8) On page 10, BMI is not defined.

==> We are sorry to make this mistake. We have defined the BMI in the revised manuscript.

(9) The added block of text on page 11 needs language editing. These new results are impressive and extremely important in addressing comments of the third reviewer but editing of the paragraph is needed. For example, I believe the third sentence down should read:

"... in various representative experiments showed that they contained ratios of EPA and DHA in the 1:1~1:2 range^{42, 43}, suggestion the ratio of EPA to DHA is critical for effects on learning and memory and other physiological process." This is just one example.

On page 15, in the first inserted block of text, there are edits needed although the meaning is clear and comments are appropriate.

==> We thank the reviewer for the thoughtful comments and the positive comments: "These new results are impressive and extremely important in addressing comments of the third reviewer". As suggested, we have made language editing for these paragraphs in the revised version.

Reviewer #3: This is an interesting paper examining acute doses of EPA on behaviour. The authors report acute EPA administration impairs memory and have done many experiments that this is dependent on 5-HT_{2c}R. While overall this is an interesting paper, the brain lipid chemistry data seems to be a odds with the literature and EPA has been given to humans at high doses (5 g per day for years) without reports of memory impairment suggesting something unique about the acute model being studied.

==> We thank the reviewer for the careful reading and the positive comments that "This is an interesting paper examining acute doses of EPA on behaviour". We also thank the reviewer for the constructive suggestions that have significantly improved the manuscript.

Major comments:

(1) *The lengthy background on CVD is a bit distracting and could be shortened. It isn't clear why the authors should focus on CVD when their work is brain-specific. There is a relatively large literature on EPA and major depression that is not commented on. See meta-analyses by Su and others suggesting benefits of EPA.*

==> We thank the reviewer for the careful reading and constructive suggestions. In the revised manuscript, we have removed several sentences on omega-3 fatty acids and CVD. Meanwhile, we added more descriptions on EPA and depression and cited the references about meta-analyses by Su and others suggesting benefits of EPA.

(2) *One of the major concerns about this study is it potential translation to humans. Numerous*

studies have been conducted with EPA and no detriments have been reported (in terms of cognitive function) if anything observational data suggests a protective effect. It seems most likely that the authors effects are model specific. Is this a result of acute dosing? Are the results seen with chronic intake?

==> We thank the reviewer for the careful reading and constructive suggestions. And we agree with the reviewer's comments that numerous studies have been conducted with EPA and no detriments have been reported (in terms of cognitive function) if anything observational data suggests a protective effect. Indeed, EPA has been reported to be benefit in the treatment of mild-to-moderate depression¹⁻⁶, attention deficit/hyperactivity disorder (ADHD)⁷ and other diseases. However, in our study, we found that EPA administration alone exhibits an unexpected detrimental impact on cognitive functions, which were observed one hour after EPA administration. From our results in analyzing the EPA level at different time point after EPA administration in Supplementary Figure 2d, we can see that EPA treatment (50 mg/kg) significantly increased the level of EPA in the hippocampus, and the increased EPA level exhibited time dependence: it increased for only one to two hours but returned to normal at three and twenty-four hours after EPA treatment (Supplementary Fig. 2d). To determine whether EPA affects the learning and memory behaviors when its level returns to normal, we performed additional experiments to test animal behaviors at two, three and twenty-four hours after EPA treatment. We found that, similar to one hour, EPA also impaired learning and memory two hours after EPA administration, but this detrimental impact disappeared three and twenty-four hours after EPA treatment (Supplementary Figure 2h-n), suggesting that the impairment of learning and memory by EPA may persist only as long as there is an elevated level of EPA in the hippocampus. Meanwhile, to examine the chronic effect, we also administered EPA (50 mg/kg) intragastrically once a day and lasted for one month. We found that EPA did not affect the fatty acids levels twenty-four hours after the last day of EPA treatment (Supplementary Fig. 2o-r), so as to the learning and memory behaviours (Supplementary Fig. 2s-z), further supporting an acute detrimental effect of EPA.

(3) Another major concern with this report is that lack of consideration of the interdependency of EPA and DHA. EPA is readily converted to DHA (Metherel AJCN 2020). How is it possible that EPA increased EPA but not DHA in the brain regions? Is this due to the acute nature of the studies? Also the large differences in DHA concentrations across brain regions (PFC, striatum and Hip) are not consistent with the literature and because they occur for all fatty acids suggests and artifact.

==> We thank the reviewer for the careful reading and constructive suggestions. And we agree with the reviewer's comments that EPA is readily converted to DHA. And in their study⁸, substantial amounts of EPA supplementation (12 weeks) did increase ¹³C-DHA but not DHA concentrations in the plasma. However, in our study, we did not find a change in the level of DHA one hour after acute EPA treatment (50 mg/kg) in the hippocampus (Supplementary Fig. 2a). The difference may be due to: (1) the sensitivity of assay method: EPA supplementation only increased less than 10% of plasma ¹³C-DHA in their study, which was not detected by the method used for measuring the DHA. (2) The duration of EPA treatment: in their study, they supplied EPA of 12 weeks, which may cause a cumulative effect that lead to an obvious

increased DHA level; but in our study, one hour after acute EPA administration was too short to convert EPA to DHA. Meanwhile, several other studies also found that EPA administration in patients only increases plasma EPA levels but not DHA levels^{6,9-11}, which were also consistent with our results.

We also agree with the reviewer's comments that the large differences in DHA concentrations across brain regions (PFC, striatum and Hip) in our results are not consistent with several investigations¹²⁻¹⁴. In our initial submission, the mice were decapitated and brain tissue was obtained in our laboratory located in Guangzhou, and because of lack of experimental condition, gas chromatography analysis can only be performed in another city Nanjing. So we guess the transportation process of the tissue may affect the results. To test this possibility, we repeated all above experiments in Nanjing and obtained similar results as our initial submission except that there were no obvious differences in EPA, DHA, AA and LA concentrations across brain regions (PFC, striatum and Hip), which was consistent with the literature¹²⁻¹⁴. So the process to transport the tissue from one city to another city may cause the large differences in DHA concentrations across brain regions. We showed these new data in the revised manuscript.

(4) How do the authors explain that EPA supplementation increased EPA selectively in the Hip (Figure 1h). Similarly speaking that fat-1 model is well known to increase brain DHA (including in the Hip – Hopperton BBI 2018; Orr JNC 201; Delpech Neuropsychopharmacology 2015). However, they do not see an increase in DHA figure S5. How is this possible? What was the background diet of these animals?

==> We thank the reviewer for the careful reading and constructive suggestions. In our study, in order to examine the effect of i.g. administration of EPA on the level of EPA in brain, we performed the gas chromatography (GC) and found that EPA treatment (50 mg/kg) remarkably elevated the level of that fatty acid in the hippocampus but not in the prefrontal cortex (PFC) or striatum (Figure 1h). To explore the possible mechanism underlying this preferential distribution in the hippocampus, we then examined the expression of fatty acid-binding protein 7 (FABP7). FABP7, also known as brain-FABP (B-FABP), was specifically expressed in various regions of the mouse brain¹⁵. FABP7 is distinguished from other FABPs by its strong affinity for omega-3 polyunsaturated fatty acids, and can uptake, transportation and storage of omega-3 polyunsaturated fatty acids¹⁵. So the expression level of FABP7 protein may be a key factor in determining the level of omega-3 polyunsaturated fatty acids in different brain region. In our study, we examined FABP7 level in three brain regions and found a higher expression in the hippocampus compared with PFC and striatum (Figure S1h). So the selective increase of EPA in the hippocampus may be due to enrichment of the FABP7 in this region.

The fat-1 transgenic mice carry the fat-1 gene that encodes an n-3 fatty-acid desaturase enzyme that converts n-6 to n-3 fatty acids¹⁶. So in this transgenic mouse line, an increased level of n-3 fatty acids such as both EPA and DHA was observed¹⁶⁻¹⁹. However, in our study, we found specific elevation of EPA in the PFC and hippocampus, with no changes in the levels of other fatty acids include DHA (Figure 2a and Figures S5c to S5e). To explore the possible reasons, we compared the difference between our and their diet and found that they maintained on a diet that is high in n-6 fatty acids (one hundred times more than that in the

normal diet)¹⁶⁻¹⁹ while our study used the standard diet, just like the control diet in their studies¹⁶⁻²⁰. As we know, the level of DHA is one hundred times more than EPA in the brain (Figure 1h and Figures S2a), which means that if we want to observe obviously elevated DHA in fat-1 mice, the mice should be fed with the food containing more n-6. So in our study, there was not enough n-6 to be converted to n-3 for leading a notable change in DHA level in the brain but it was enough for elevating EPA. And this explanation was also consistent with the conclusions from Corinne Joffre et al's study²¹, which indicated that DHA levels in brain were differently affected by dietary approaches.

References

1. Lin, P. Y. & Su, K. P. A meta-analytic review of double-blind, placebo-controlled trials of antidepressant efficacy of omega-3 fatty acids. *J Clin Psychiatry*. **68**, 1056-1061 (2007).
2. Lin, P., Huang, S. & Su, K. A Meta-Analytic Review of Polyunsaturated Fatty Acid Compositions in Patients with Depression. *Biol. Psychiat*. **68**, 140-147 (2010).
3. Lin, P. Y., Chang, C. H., Chong, M. F., Chen, H. & Su, K. P. Polyunsaturated Fatty Acids in Perinatal Depression: A Systematic Review and Meta-analysis. *Biol Psychiatry*. **82**, 560-569 (2017).
4. Frangou, S., Lewis, M. & McCrone, P. Efficacy of ethyl-eicosapentaenoic acid in bipolar depression: randomised double-blind placebo-controlled study. *Br J Psychiatry*. **188**, 46-50 (2006).
5. Peet, M. & Horrobin, D. F. A Dose-Ranging Study of the Effects of Ethyl-Eicosapentaenoate in Patients With Ongoing Depression Despite Apparently Adequate Treatment With Standard Drugs. *Archives of General Psychiatry*. **59**, 913-919 (2002).
6. Mischoulon, D. et al. A Double-Blind, Randomized Controlled Trial of Ethyl-Eicosapentaenoate for Major Depressive Disorder. *The Journal of Clinical Psychiatry*. **70**, 1636-1644 (2009).
7. Stevens, L. J. et al. Essential fatty acid metabolism in boys with attention-deficit hyperactivity disorder. *Am. J. Clin. Nutr.* **62**, 761-768 (1995).
8. Metherel, A. H., Irfan, M., Klingel, S. L., Mutch, D. M. & Bazinet, R. P. Compound-specific isotope analysis reveals no retroconversion of DHA to EPA but substantial conversion of EPA to DHA following supplementation: a randomized control trial. *The American Journal of Clinical Nutrition*. **110**, 823-831 (2019).
9. Chang, J. P. et al. High-dose eicosapentaenoic acid (EPA) improves attention and vigilance in children and adolescents with attention deficit hyperactivity disorder (ADHD) and low endogenous EPA levels. *Transl. Psychiat*. **9**, (2019).
10. Arterburn, L. M., Hall, E. B. & Oken, H. Distribution, interconversion, and dose response of n-3 fatty acids in humans. *Am. J. Clin. Nutr.* **83**, 1467S-1476S (2006).
11. Nestel, P. et al. The n-3 fatty acids eicosapentaenoic acid and docosahexaenoic acid increase systemic arterial compliance in humans. *Am. J. Clin. Nutr.* **76**, 326-330 (2002).
12. Favreliere, S., Barrier, L., Durand, G., Chalon, S. & Tallineau, C. Chronic dietary n-3 polyunsaturated fatty acids deficiency affects the fatty acid composition of plasmamylethanolamine and phosphatidylethanolamine differently in rat frontal cortex, striatum,

- and cerebellum. *Lipids*. **33**, 401-407 (1998).
13. Xiao, Y., Huang, Y. & Chen, Z. Y. Distribution, depletion and recovery of docosahexaenoic acid are region-specific in rat brain. *Br J Nutr*. **94**, 544-550 (2005).
 14. Carrie, I., Clement, M., de Javel, D., Frances, H. & Bourre, J. M. Specific phospholipid fatty acid composition of brain regions in mice. Effects of n-3 polyunsaturated fatty acid deficiency and phospholipid supplementation. *J. Lipid Res*. **41**, 465-472 (2000).
 15. Furuhashi, M. & Hotamisligil, G. S. Fatty acid-binding proteins: role in metabolic diseases and potential as drug targets. *Nat. Rev. Drug Discov*. **7**, 489-503 (2008).
 16. Kang, J. X., Wang, J., Wu, L. & Kang, Z. B. Transgenic mice: fat-1 mice convert n-6 to n-3 fatty acids. *Nature*. **427**, 504 (2004).
 17. Orr, S. K. et al. Unesterified docosahexaenoic acid is protective in neuroinflammation. *J. Neurochem*. **127**, 378-393 (2013).
 18. Delpech, J. C. et al. Transgenic increase in n-3/n-6 fatty acid ratio protects against cognitive deficits induced by an immune challenge through decrease of neuroinflammation. *Neuropsychopharmacol*. **40**, 525-536 (2015).
 19. Hopperton, K. E., Trépanier, M., James, N. C. E., Chouinard-Watkins, R. & Bazinet, R. P. Fish oil feeding attenuates neuroinflammatory gene expression without concomitant changes in brain eicosanoids and docosanoids in a mouse model of Alzheimer's disease. *Brain, Behavior, and Immunity*. **69**, 74-90 (2018).
 20. Jašarević, E., Hecht, P. M., Fritsche, K. L., Beversdorf, D. Q. & Geary, D. C. Dissociable effects of dorsal and ventral hippocampal DHA content on spatial learning and anxiety-like behavior. *Neurobiol. Learn. Mem*. **116**, 59-68 (2014).
 21. Joffre, C. et al. Modulation of brain PUFA content in different experimental models of mice. *Prostaglandins, Leukotrienes and Essential Fatty Acids*. **114**, 1-10 (2016).

Reviewers' Comments:

Reviewer #2:

Remarks to the Author:

The authors made important revisions to the manuscript including the use of more specific language in the abstract and introduction. They corrected some errors in wording and provide more description (and precise articulation of procedures) relating to the duration of effects on synaptic plasticity and learning.

In general the experimental designs associated with LTP and learning experiments are now well described – they provide timing of experiments with the SB antagonist and also better describe the results with the GABA receptor antagonist. These are important components of the study and with the revisions the results better support the conclusions reached. The large block of text at the end of the Results section is well written. The concerns of this reviewer have been addressed.

Small items.

On page 4, 7 lines from the bottom, it would read better if they stated "...once a day for a period of one month and...". On the same page beginning 5 lines from the bottom it should be revised....to clearly state the treatment did not affect learning and memory with this timing "...so as to the..." does not make this point. The figures pertaining to these statements are excellent.

Reviewer #3:

Remarks to the Author:

The authors have a robust effect in their model and have done extensive work evaluating the mechanism of action. The authors have largely addressed my concerns by redoing an experiment, editing out sections on cardiovascular disease and pointing out differences in study designs thus questioning if results from other studies can be extrapolated to their findings. However, they have not really addressed my concern about generalization to humans.

We now have data from over a decade of large high quality RCTs Bhatt et al., N Engl J Med (>8000 subjects) . 2019 Jan 3;380(1):11-22; Yokoyama et al., Lancet 2007 (>18,000 subjects) with high purity EPA. We have meta-analyses of brain outcomes and side effects. The FDA has approved an EPA ethyl ester for triglyceride lowering and the prevention of heart disease.

Meta-analyses of clinical studies <https://pubmed.ncbi.nlm.nih.gov/32070694/> conclude "Subgroup analyses identified beneficial effects of eicosapentaenoic acid (EPA)-rich but not docosahexaenoic acid (DHA)-rich formulations in the domains of long-term memory, working memory and problem solving and a tendency towards beneficial effects in clinical rather than non-clinical populations. " - essentially the opposite of this preclinical study.

There is also a preclinical literature that seems at odds with the memory impairment <https://pubmed.ncbi.nlm.nih.gov/20621517/>, albeit study designs are very different.

In their reply the authors did little to address my concern that their effect was model specific and might not apply to humans. The authors are arguing, perhaps correctly, that their effects are due to acute EPA. This has not been reported in the clinical literature a) there are no adverse reports upon beginning the studies and b) this could easily be tested in humans.

Several statements in the revision do not appear to be accurate or supported by the citations given.

"In contrast to EPA, marine fish oil, a rich source of omega-3 fatty acids, especially EPA and DHA, has no reported adverse effects on learning or memory; the same is true of combined supplementation with EPA and DHA" – see the meta-analyses above.

"Linoleic acid and α -linolenic acid (the precursors of EPA and DHA) are not synthesized de novo by mammals, and a balanced diet containing appropriate amounts of these precursors is necessary to maintain sufficient brain levels of EPA and DHA" – linoleic acid is not a precursor to EPA or DHA, but rather arachidonic acid.

"It has been found that DHA alone can attenuate age-related cognitive decline²¹⁻²³. However, recent clinical studies reported a negative result when DHA was combined with EPA at a ratio of 1:150³⁵⁸, which naturally raises the question of whether EPA exerts an effect on cognitive function opposite to the effect of DHA." – citation 21 was null. Citation 22, a industry study, changed their primary outcome after registration. 23 is not a clinical study about DHA. And notwithstanding my previous comments, these are chronic studies. How does citation 50, a multi year study apply to the authors acute findings?

"Furthermore, when we compared the composition of various representative samples of these commonly consumed fish oils as well as EPA and DHA supplements, we found that they contained different EPA and DHA in different ratios^{45,46}, ranging from 1:1 to 1:2 in different studies." – these are largely cardiovascular studies. As I mentioned there are major studies examining just EPA.

"For example, seven double-blind randomized controlled trials have investigated the effect of omega-3 fatty acids supplementation on depression and found a significant beneficial effect of EPA plus DHA on bipolar depression^{6,7}. – citation 6 is not a study, but a protocol and we are awaiting the results of this study NCT02166424 and 7 is just one study. See this meta analysis <https://pubmed.ncbi.nlm.nih.gov/27103682/>

A point-by-point reply to the reviewers follows:

Reviewer #2: The authors made important revisions to the manuscript including the use of more specific language in the abstract and introduction. They corrected some errors in wording and provide more description (and precise articulation of procedures) relating to the duration of effects on synaptic plasticity and learning.

In general the experimental designs associated with LTP and learning experiments are now well described – they provide timing of experiments with the SB antagonist and also better describe the results with the GABA receptor antagonist. These are important components of the study and with the revisions the results better support the conclusions reached. The large block of text at the end of the Results section is well written. The concerns of this reviewer have been addressed.

==> We thank the reviewer for the careful reading and the positive comments that “the experimental designs associated with LTP and learning experiments are now well described” and “The large block of text at the end of the Results section is well written” and “The concerns of this reviewer have been addressed”. We also thank the reviewer for the constructive suggestions that have significantly improved the manuscript.

Minor comments:

On page 4, 7 lines from the bottom, it would read better if they stated “...once a day for a period of one month and...”. On the same page beginning 5 lines from the bottom it should be revised...to clearly state the treatment did not affect learning and memory with this timing “...so as to the....” does not make this point. The figures pertaining to these statements are excellent.

==> Good suggestions. We have rewritten this part as suggested in the revised manuscript.

Reviewer #3: The authors have a robust effect in their model and have done extensive work evaluating the mechanism of action. The authors have largely addressed my concerns by redoing an experiment, editing out sections on cardiovascular disease and pointing out differences in study designs thus questioning if results from other studies can be extrapolated to their findings. However, they have not really addressed my concern about generalization to humans.

==> We thank the reviewer for the careful reading and the positive comments that “The authors have a robust effect in their model and have done extensive work evaluating the mechanism of action. The authors have largely addressed my concerns by redoing an experiment, editing out sections on cardiovascular disease and pointing out differences in study designs thus questioning if results from other studies can be extrapolated to their findings”. We also thank the reviewer for the constructive suggestions that have significantly improved the manuscript.

Major comments:

(1) *We now have data from over a decade of large high quality RCTs Bhatt et al., N Engl J Med (>8000 subjects) . 2019 Jan 3;380(1):11-22; Yokoyama et al., Lancet 2007 (>18,000 subjects) with high purity EPA. We have meta-analyses of brain outcomes and side effects. The FDA has approved an EPA ethyl ester for triglyceride lowering and the prevention of heart disease.*

Meta-analyses of clinical studies <https://pubmed.ncbi.nlm.nih.gov/32070694/> conclude "Subgroup analyses identified beneficial effects of eicosapentaenoic acid (EPA)-rich but not docosahexaenoic acid (DHA)-rich formulations in the domains of long-term memory, working memory and problem solving and a tendency towards beneficial effects in clinical rather than non-clinical populations. " - essentially the opposite of this preclinical study. There is also a preclinical literature that seems at odds with the memory impairment <https://pubmed.ncbi.nlm.nih.gov/20621517/>, albeit study designs are very different.

In their reply the authors did little to address my concern that their effect was model specific and might not apply to humans. The authors are arguing, perhaps correctly, that their effects are due to acute EPA. This has not been reported in the clinical literature a) there are no adverse reports upon beginning the studies and b) this could easily be tested in humans.

==> We thank the reviewer for the careful reading and constructive suggestions. We agree with the reviewer's comments that we did little to address the reviewer's concern about generalization to humans. Indeed, the finding of our study that impaired role of EPA on cognition might not apply to human right now as we did not test it in humans. So we changed our title into "Acute EPA-induced learning and memory impairment in mice is prevented by DHA" to better describe our study. We also added the following discussion in our revised manuscript: "In contrast, recent meta-analyses of clinical studies identified beneficial effects for treatment with EPA-rich formulations in the domains of long-term memory, working memory and problem solving¹. However, these studies were performed in a chronic manner but there was a lack of acute effect of EPA on human cognition. So future investigations should take the time window of cognition detection into consideration". Meanwhile, we have modified some descriptions by prescribing a limit to an acute manner in EPA administration and tone down the claims regarding the relevance to humans.

Minor comments:

Several statements in the revision do not appear to be accurate or supported by the citations given.

(1) *"In contrast to EPA, marine fish oil, a rich source of omega-3 fatty acids, especially EPA and DHA, has no reported adverse effects on learning or memory; the same is true of combined supplementation with EPA and DHA" – see the meta-analyses above.*

==> We thank the reviewer for the careful reading and constructive suggestions. We have changed this statement in our revised manuscript: "Marine fish oil, a rich source of omega-3

fatty acids, especially EPA and DHA, has no reported adverse effects on learning or memory; the same is true of combined supplementation with EPA and DHA ^{1,2}” and cited the meta-analyses of clinical studies that the reviewer mentioned.

(2) “Linoleic acid and α -linolenic acid (the precursors of EPA and DHA) are not synthesized de novo by mammals, and a balanced diet containing appropriate amounts of these precursors is necessary to maintain sufficient brain levels of EPA and DHA” – linoleic acid is not a precursor to EPA or DHA, but rather arachidonic acid.

==> We thank the reviewer for the careful reading and constructive suggestions. We are sorry to make this mistake. We have changed the description in the revised manuscript.

(3) “It has been found that DHA alone can attenuate age-related cognitive decline²¹⁻²³. However, recent clinical studies reported a negative result when DHA was combined with EPA at a ratio of 1:1⁵⁰, which naturally raises the question of whether EPA exerts an effect on cognitive function opposite to the effect of DHA.” – citation 21 was null. Citation 22, a industry study, changed their primary outcome after registration. 23 is not a clinical study about DHA. And notwithstanding my previous comments, these are chronic studies. How does citation 50, a multi year study apply to the authors acute findings?

==> We thank the reviewer for the careful reading and constructive suggestions. We have deleted the above sentences in our revised manuscript.

(4) “Furthermore, when we compared the composition of various representative samples of these commonly consumed fish oils as well as EPA and DHA supplements, we found that they contained different EPA and DHA in different ratios^{45,46}, ranging from 1:1 to 1:2 in different studies.” – these are largely cardiovascular studies. As I mentioned there are major studies examining just EPA.

==> We thank the reviewer for the careful reading and constructive suggestions. We have replaced the above cited references with current ones studying the EPA on cognition.

(5) “For example, seven double-blind randomized controlled trials have investigated the effect of omega-3 fatty acids supplementation on depression and found a significant beneficial effect of EPA plus DHA on bipolar depression^{6,7}.” – citation 6 is not a study, but a protocol and we are awaiting the results of this study NCT02166424 and 7 is just one study . See this meta analysis <https://pubmed.ncbi.nlm.nih.gov/27103682/>

==> We thank the reviewer for the careful reading and constructive suggestions. We have rewritten this statement in our revised manuscript: “For example, a meta-analysis performed in 35 double-blind RCTs including 6665 participants receiving omega-3 HUFAs and 4373 participants receiving placebo found a positive role of omega-3 fatty acids supplementation on depression ³”.

References

1. Emery, S. et al. Omega-3 and its domain-specific effects on cognitive test performance in youths: A meta-analysis. *Neuroscience & Biobehavioral Reviews*. **112**, 420-436 (2020).
2. Chew, E. Y. et al. Effect of Omega-3 Fatty Acids, Lutein/Zeaxanthin, or Other Nutrient Supplementation on Cognitive Function. *JAMA*. **314**, 791 (2015).
3. Hallahan, B. et al. Efficacy of omega-3 highly unsaturated fatty acids in the treatment of depression. *Br J Psychiatry*. **209**, 192-201 (2016).